# Unsupervised Manifold Alignment with Joint Multidimensional Scaling

**Dexiong Chen, Bowen Fan, Carlos Oliver & Karsten Borgwardt** [*]
Department of Biosystems Science and Engineering, ETH Zürich, Switzerland
SIB, Swiss Institute of Bioinformatics, Switzerland
`{firstname.lastname}@bsse.ethz.ch`

## Abstract

We introduce Joint Multidimensional Scaling, a novel approach for unsupervised manifold alignment, which maps datasets from two different domains, without any known correspondences between data instances across the datasets, to a common low-dimensional Euclidean space. Our approach integrates Multidimensional Scaling (MDS) and Wasserstein Procrustes analysis into a joint optimization problem to simultaneously generate isometric embeddings of data and learn correspondences between instances from two different datasets, while only requiring intra-dataset pairwise dissimilarities as input. This unique characteristic makes our approach applicable to datasets without access to the input features, such as solving the inexact graph matching problem. We propose an alternating optimization scheme to solve the problem that can fully benefit from the optimization techniques for MDS and Wasserstein Procrustes. We demonstrate the effectiveness of our approach in several applications, including joint visualization of two datasets, unsupervised heterogeneous domain adaptation, graph matching, and protein structure alignment. The implementation of our work is available at https://github.com/BorgwardtLab/JointMDS.

## 1 Introduction

Many problems in machine learning require joint visual exploration and manipulation of multiple datasets from different (heterogeneous) domains, which is generally a preferable first step prior to any further data analysis. These different data domains may consist of measurements for the same samples obtained with different methods or technologies, such as single-cell multi-omics data in bioinformatics (Demetci et al., 2022; Liu et al., 2019; Cao & Gao, 2022). Alternatively, the data could be comprised of different datasets of similar objects, such as word spaces of different languages in natural language modeling (Alvarez-Melis et al., 2019; Grave et al., 2019), or graphs representing related objects such as disease-procedure recommendation in biomedicine (Xu et al., 2019b). There are two main challenges in joint exploration of multiple datasets. First, the data from the heterogeneous domains may be high-dimensional or may not possess input features but rather only dissimilarities between them. Second, the correspondences between data instances across different domains may not be known *a priori*. We propose in this work to tackle both issues simultaneously while making few assumptions on the data modality.

To address the first challenge, for several decades many dimensionality reduction methods have been proposed to provide lower-dimensional embeddings of data. Among them, multidimensional scaling (MDS) (Borg & Groenen, 2005) and its extensions (Tenenbaum et al., 2000; Chen & Buja, 2009) are widely used ones. They generate low-dimensional data embeddings while preserving the local and global information about its manifold structure. One of the key characteristics of MDS is the fact that it only requires pairwise (dis)similarities as input rather than specific data features, which makes it applicable to problems whose data does not have access to the input features, such as graph node embedding learning (Gansner et al., 2004). However, when it comes to dealing with multiple datasets from different domains at the same time, the subspaces learned by MDS for different datasets are not naturally aligned, making it not directly applicable for a joint exploration.

---

[*]Dexiong Chen and Bowen Fan contributed equally.

One well-known method for aligning data instances from different spaces is Procrustes analysis. When used together with dimensionality reduction, it results in a manifold alignment method (Wang & Mahadevan, 2008; Kohli et al., 2021; Lin et al., 2021). However, these approaches require prior knowledge about the correspondences between data instances across the domains, which limits their applicability in many real-world applications where this information is hard or expensive to obtain. Unsupervised manifold alignment approaches (Wang & Mahadevan, 2009; Cui et al., 2014) have been proposed to overcome this limitation by aligning the underlying manifold structures of two datasets with unknown correspondences while projecting data onto a common low-dimensional space.

In this work, we propose to combine MDS with the idea of unsupervised manifold alignment to simultaneously embed data instances from two domains without known correspondences to a common low-dimensional space, while only requiring intra-dataset dissimilarities. We formulate the problem as a joint optimization problem, where we integrate the stress functions for each dataset that measure the distance deviations and adopt the idea of Wasserstein Procrustes analysis (Alvarez-Melis et al., 2019) to align the embedded data instances from two datasets in a fully unsupervised manner. We propose to solve the resulting optimization problem through an alternating optimization strategy, resulting in an algorithm that can benefit from the optimization techniques for solving each individual sub-problem. Our approach, named Joint MDS, allows recovering the correspondences between instances across domains while also producing aligned low-dimensional embeddings for data from both domains, which is the main advantage compared to Gromov-Wasserstein (GW) optimal transport (Mémoli, 2011; Yan et al., 2018) for only correspondence finding. We show the effectiveness of joint MDS in several machine learning applications, including joint visualization of two datasets, unsupervised heterogeneous domain adaptation, graph matching, and protein structure alignment.

## 2 RELATED WORK

We present here the work most related to ours, namely MDS, unsupervised manifold alignment and optimal transport (OT) for correspondence finding.

**Multidimensional scaling and extensions**  MDS is one of the most commonly used dimensionality reduction methods that only require pairwise (dis)similarities between data instances as input. Classical MDS (Torgerson, 1965) was introduced under the assumption that the dissimilarity is an Euclidean distance, which has an analytic solution via SVD. As an extension of classic MDS, metric MDS consists in learning low-dimensional embeddings that preserve any dissimilarity by minimizing a *stress* function. Several extensions of MDS have also been proposed for various practical reasons, such as non-metric MDS (Agarwal et al., 2007), Isomap (Tenenbaum et al., 2000), local MDS (Chen & Buja, 2009) and so on. MDS has also been used for graph drawing (Gansner et al., 2004) by producing node embeddings using shortest path distances on the graph. Our approach can be seen as an important extension of MDS to work with multiple datasets.

**Unsupervised manifold alignment**  While (semi-)supervised manifold alignment methods (Ham et al., 2005; Wang & Mahadevan, 2008; Shon et al., 2005) require at least partial information about the correspondence across domains, unsupervised manifold alignment learns the correspondence directly from the underlying structures of the data. One of the earliest works for unsupervised manifold alignment was presented in (Wang & Mahadevan, 2009), where a similarity metric based on the permutation of the local geometry was used to find cross-domain corresponding instances followed by a non-linear dimensionality reduction. A similar approach was also adopted in (Tuia et al., 2014) with a graph-based similarity metric. A more generalized framework, named GUMA (Cui et al., 2014), was proposed as an optimization problem with three complex terms to project data instances via a linear transformation and match them simultaneously. As an extension of (Cui et al., 2014), UnionCom (Cao et al., 2020) introduced geodesic distance matching instead of the kernel matrices, to deal with multi-model data in bioinformatics in particular. Additionally, generative adversarial networks and the maximum mean discrepancy have also been used to find correspondences jointly with dimensionality reduction for unsupervised manifold alignment (Amodio & Krishnaswamy, 2018; Liu et al., 2019). Our approach differs from these previous approaches as it makes few assumptions on the data modality but only requires intra-dataset dissimilarities as input.

**Optimal transport for correspondence finding**  OT (Peyré et al., 2019) is a powerful and flexible approach to compare two distributions, and has a strong theoretical foundation. It can find a soft-correspondence mapping between two sets of samples without any supervision. With the

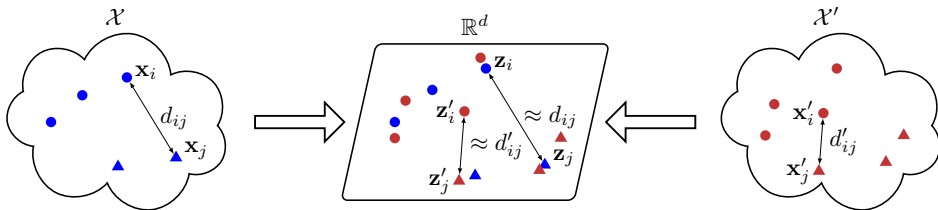

Figure 1: Joint Multidimensional Scaling maps data from two domains $\mathcal{X}$ and $\mathcal{X}'$ to a common low-dimensional space $\mathbb{R}^d$ while preserving the pairwise intra-domain dissimilarities.

recent advances in computational algorithms for OT (Cuturi, 2013), it has become increasingly popular in various machine learning fields. However, one major drawback of OT is that it was originally designed to compare two distributions in the same metric space: the cost function cannot be properly defined between samples in different spaces. This limitation makes it unsuitable for tasks involving data from different domains. As an important extension, the Gromov-Wasserstein (Mémoli, 2011) distance has been proposed to overcome this limitation, by comparing two pairwise intra-domain (dis)similarity matrices directly. This modification makes the GW distance invariant to many geometric transformations on the data, such as rotation or translation. GW has thus been successfully applied to several tasks involving datasets from heterogeneous domains, such as heterogeneous domain adaptation (HDA) (Yan et al., 2018; Titouan et al., 2020), graph matching and partitioning (Xu et al., 2019b; Vayer et al., 2019), and integration of single-cell multi-omics data (Demetci et al., 2022). As an alternative of GW, OT methods with global invariances have also been recently proposed for unsupervised word translation tasks in NLP (Alvarez-Melis et al., 2019; Grave et al., 2019), and for more general purpose (Jin et al., 2021). In addition to discovering correspondences, our approach can additionally produce low-dimensional embeddings of data from both domains, which could be useful for visualization and further analysis.

## 3 BACKGROUND

In this section, we revisit the metric MDS problem and Wasserstein Procrustes, two main building blocks of our method.

**Notation**  We denote by $\mathbf{D} \in \mathbb{R}^{n \times n}$ the pairwise dissimilarities of a dataset of $n$ elements (in an unknown space $\mathcal{X}$) and by $d : \mathbb{R}^d \times \mathbb{R}^d \to \mathbb{R}_+$ a distance function which is typically the Euclidean distance. For two sets of data points $\mathbf{Z} \in \mathbb{R}^{n \times d}$ and $\mathbf{Z}' \in \mathbb{R}^{n' \times d}$ of respective $n$ and $n'$ elements in $\mathbb{R}^d$, we denote by $d(\mathbf{Z}, \mathbf{Z}')$ the pairwise distance matrix in $\mathbb{R}^{n \times n'}$ such that its $(i, j)$-th entry is equal to $d(\mathbf{Z}_i, \mathbf{Z}'_j)$. We denote by $\| \cdot \|_F$ the Frobenius norm of matrix and $\langle \cdot, \cdot \rangle_F$ the associated inner-product. We denote by $\Delta_n$ the probability simplex with $n$ bins $\Delta_n := \{\mathbf{a} \in \mathbb{R}^n_+ \mid \sum_{j=1}^n \mathbf{a}_j = 1\}$.

**Metric multidimensional scaling**  Given a dissimilarity matrix $\mathbf{D} \in \mathbb{R}^{n \times n}$ with $(i, j)$-th entry denoted as $d_{ij}$, representing a set of $n$ samples in an unknown space $\mathcal{X}$, MDS consists of finding $n$ coordinates $\mathbf{Z} := (\mathbf{z}_1, \ldots, \mathbf{z}_n)^\top \in \mathbb{R}^{n \times d}$ that best preserves the pairwise dissimilarities by solving the following weighted stress minimization problem:

$$\min_{\mathbf{Z} \in \mathbb{R}^{n \times d}} \text{stress}(\mathbf{Z}, \mathbf{D}, \mathbf{W}) := \sum_{i,j=1}^n w_{ij}(d_{ij} - d(\mathbf{z}_i, \mathbf{z}_j))^2, \tag{1}$$

for a given weight matrix $\mathbf{W} := (w_{ij})_{i,j=1,\ldots,n} \in \mathbb{R}^{n \times n}_+$. Despite being a non-convex problem, it can be efficiently solved by an iterative majorization algorithm, named SMACOF (scaling by majorizing a complicated function) (Borg & Groenen, 2005). This algorithm could be used individually to map two datasets to two low-dimensional Euclidean spaces. However, both the alignment of the two subspaces and the correspondences between data instances are not known *a priori*. It is thus meaningless to work jointly with the embeddings of both datasets directly.

**Wasserstein Procrustes**  OT is a widely used approach for finding correspondences between feature vectors from the same space in an unsupervised fashion. Its strong theoretical foundations and fast algorithms make it a natural candidate to align the distributions of the embedded data obtained with MDS. Specifically, let $\mathbf{a} \in \Delta_n$ and $\mathbf{b} \in \Delta_{n'}$ be the weights of the discrete measures $\sum_i \mathbf{a}_i \delta_{\mathbf{z}_i}$ and $\sum_j \mathbf{b}_j \delta_{\mathbf{z}'_j}$ with respective location $\mathbf{Z} := (\mathbf{z}_1, \ldots, \mathbf{z}_n)^\top \in \mathbb{R}^{n \times d}$ and $\mathbf{Z}' := (\mathbf{z}'_1, \ldots, \mathbf{z}'_{n'})^\top \in \mathbb{R}^{n' \times d}$.

For a cost function $c : \mathbb{R}^d \times \mathbb{R}^d \to \mathbb{R}_+$, Kantorovich's formulation of the OT problem between the two discrete measures is defined as

$$\min_{\mathbf{P} \in \Pi(\mathbf{a}, \mathbf{b})} \langle \mathbf{P}, \mathbf{C} \rangle_F, \tag{2}$$

where $\mathbf{C} \in \mathbb{R}^{n \times n'}$ represents the pairwise costs whose entries are $\mathbf{C}_{ij} := c(\mathbf{z}_i, \mathbf{z}'_j)$ and $\Pi(\mathbf{a}, \mathbf{b})$ is the polytope of the admissible couplings between $\mathbf{a}$ and $\mathbf{b}$:

$$\Pi(\mathbf{a}, \mathbf{b}) := \{\mathbf{P} \in \mathbb{R}_+^{n \times n'} \,|\, \mathbf{P}\mathbf{1} = \mathbf{a}, \mathbf{P}^\top \mathbf{1} = \mathbf{b}\}. \tag{3}$$

In practice, $\mathbf{a}$ and $\mathbf{b}$ are uniform measures as we consider the mass to be evenly distributed among the data points. In particular, if $c$ is the squared Euclidean distance such that $c(\mathbf{z}, \mathbf{z}') = \|\mathbf{z} - \mathbf{z}'\|_2^2$, the solution of Eq. (2) defines a squared distance on the set of discrete measures, called the squared Wasserstein distance, which we denote by $W_2^2(\mathbf{Z}, \mathbf{Z}')$ below.

Naive application of OT to find correspondences between the embedded data obtained with two MDSs for two datasets can easily fail since the two spaces of $\mathbf{Z}$ and $\mathbf{Z}'$ are not coherent, or in other words, there is no meaningful notion of distance between them. In order to jointly align the two spaces and find the correspondences, Alvarez-Melis et al. (2019) propose to jointly optimize a linear transformation $f$ from a pre-defined invariance class $\mathcal{F}$:

$$\min_{f \in \mathcal{F}} W_2^2(f(\mathbf{Z}), \mathbf{Z}') = \min_{\mathbf{P} \in \Pi(\mathbf{a}, \mathbf{b})} \min_{f \in \mathcal{F}} \langle \mathbf{P}, d^2(f(\mathbf{Z}), \mathbf{Z}') \rangle_F. \tag{4}$$

They provide solutions of this problem for invariances defined by linear operators with a bounded norm $\mathcal{F} := \{O \in \mathbb{R}^{d \times d} \,|\, \|O\|_p \leq k_p\}$, where $\|\cdot\|_p$ is the Schatten $\ell_p$-norm. In particular, when $p = \infty$ and $k_p = 1$, we have $\mathcal{F} = \mathcal{O}_d$, the orthogonal group of matrices in $\mathbb{R}^{d \times d}$, and one recovers the Wasserstein Procrustes problem (Grave et al., 2019).

## 4 JOINT MULTIDIMENSIONAL SCALING PROBLEM

We introduce our new joint MDS problem which can jointly embed two datasets in a common space.

### 4.1 CORRESPONDENCE DISCOVERY WITH JOINT MULTIDIMENSIONAL SCALING

The Joint MDS problem consists of jointly mapping data points from two different domains into a common Euclidean space. Following the discussion in Section 3, the two spaces obtained with two individual MDSs for each dataset are not necessarily aligned and the correspondences between data instances are unknown. To address both issues, we leverage Wasserstein Procrustes to algin the embedded data from two domains.

Specifically, we combine the stress minimization and the Wasserstein Procrustes to define the overall objective function for our Joint MDS problem. Given two matrices $\mathbf{D} \in \mathbb{R}^{n \times n}$ and $\mathbf{D}' \in \mathbb{R}^{n' \times n'}$ representing the pairwise dissimilarities of two collections, we formulate the Joint MDS as the following optimization problem:

$$\min_{\substack{\mathbf{Z} \in \mathbb{R}^{n \times d}, \mathbf{Z}' \in \mathbb{R}^{n' \times d} \\ \mathbf{P} \in \Pi(\mathbf{a}, \mathbf{b}), \mathbf{O} \in \mathcal{O}_d}} \operatorname{stress}(\mathbf{Z}, \mathbf{D}, \mathbf{W}) + \operatorname{stress}(\mathbf{Z}', \mathbf{D}', \mathbf{W}') + 2\lambda \langle \mathbf{P}, d^2(\mathbf{Z}\mathbf{O}, \mathbf{Z}') \rangle_F, \tag{5}$$

where $\mathbf{W}$ and $\mathbf{W}'$ are weights for the pairwise dissimilarities generally equal to $1/n^2$ and $1/n'^2$, and $\mathbf{a}$ and $\mathbf{b}$ refer to the sample weights as defined above. This objective function exhibits a simple interpretation: the two stress function terms measure the distance deviation for each domain while the last term measures the correspondences between instances from the two domains. It is worth noting that the stress functions are rotation-invariant, *i.e.* if $\mathbf{Z}$ and $\mathbf{Z}'$ are solutions respectively minimizing the two stress functions, their rotations are also solutions. Thus, using the Wasserstein Procrustes with global geometric invariances (in $\mathcal{O}_d$) instead of the original OT could find better correspondences. An illustration of Joint MDS is shown in Figure 1.

### 4.2 CHOICE OF THE DISSIMILARITIES

In general, any distance could be used to compute the dissimilarities if they are not provided directly with the dataset. For instance, the Euclidean distance is a natural choice. As an extension of the

Euclidean distance in Riemannian geometry, the geodesic distance (Tenenbaum et al., 2000) often captures better the local geometric structure of the data distribution. However, the geodesic distance is generally hard to compute exactly as it requires full knowledge about the data manifold. Fortunately, it can be efficiently estimated approximately by constructing a $k$-nearest neighbor graph and computing the shortest path distance on the graph. Similar to Isomap (Tenenbaum et al., 2000), a variant of MDS computed on the geodesic distance, we also observe better performance for joint MDS when using geodesic distances, especially in unsupervised heterogeneous domain adaptation tasks.

## 4.3 Optimization

The above optimization problem is non-convex, which is hard to solve in general. Here, we propose an alternating optimization scheme by solving two subproblems that have already been studied previously. Specifically, we show that when $\mathbf{P}$ and $\mathbf{O}$ are fixed, the Joint MDS problem (5) becomes a simple weighted MDS problem. On the other hand, when fixing $\mathbf{Z}$ and $\mathbf{Z}'$, the problem (5) is reduced to a Wasserstein Procrustes problem. As a consequence, we can easily leverage optimization techniques developed for each of the sub-problems respectively.

**Weighted MDS problem** We first observe that the distance is invariant to the orthogonal transformations from $\mathcal{O}_d$. Thus, for fixed $\mathbf{P}$ and $\mathbf{O}$, the problem amounts to minimizing the following *stress* function (with a change of variable $\mathbf{Z} = \mathbf{Z}\mathbf{O}$):

$$\min_{\mathbf{Z}\in\mathbb{R}^{n\times d},\mathbf{Z}'\in\mathbb{R}^{n'\times d}} \sum_{i,j=1}^{n} w_{ij}(d_{ij} - d(\mathbf{z}_i, \mathbf{z}_j))^2 + \sum_{i',j'=1}^{n'} w'_{i'j'}(d'_{i'j'} - d(\mathbf{z}'_{i'}, \mathbf{z}'_{j'}))^2 + 2\lambda\langle \mathbf{P}, d^2(\mathbf{Z}, \mathbf{Z}')\rangle,$$

(6)

where the last term can be rewritten as

$$2\lambda\langle \mathbf{P}, d^2(\mathbf{Z}, \mathbf{Z}')\rangle = \sum_{i=1}^{n}\sum_{j'=1}^{n'} \lambda\mathbf{P}_{ij'}(0 - d(\mathbf{z}_i, \mathbf{z}'_{j'}))^2 + \sum_{i'=1}^{n'}\sum_{j=1}^{n} \lambda\mathbf{P}^\top_{i'j}(0 - d(\mathbf{z}'_{i'}, \mathbf{z}_j))^2.$$

This is equivalent to solving the MDS problem with a stress function $\mathrm{stress}(\tilde{\mathbf{Z}}, \tilde{\mathbf{D}}, \tilde{\mathbf{W}})$ where

$$\tilde{\mathbf{Z}} := \begin{bmatrix} \mathbf{Z} \\ \mathbf{Z}' \end{bmatrix}, \quad \tilde{\mathbf{D}} := \begin{bmatrix} \mathbf{D} & 0 \\ 0 & \mathbf{D}' \end{bmatrix}, \quad \tilde{\mathbf{W}} := \begin{bmatrix} \mathbf{W} & \lambda\mathbf{P} \\ \lambda\mathbf{P}^\top & \mathbf{W}' \end{bmatrix}.$$

(7)

This problem can be solved with the SMACOF algorithm, which simply relies on the majorization theory without making any assumption on whether $\tilde{\mathbf{D}}$ has a distance structure. The majorization principle essentially consists in minimizing a more manageable surrogate function $g(\mathbf{X}, \mathbf{Z})$ rather than directly minimizing the original complicated function $h(\mathbf{X}) := \mathrm{stress}(\mathbf{X}, \mathbf{D}, \mathbf{W})$. This surrogate function $g(\mathbf{X}, \mathbf{Z})$ is required to (i) be a majorizing function of $h(\mathbf{X})$, *i.e.* $h(\mathbf{X}) \leq g(\mathbf{X}, \mathbf{Z})$ for any $\mathbf{X} \in \mathbb{R}^{n\times d}$, and (ii) touch the surface at the so-called supporting point $\mathbf{Z}$, *i.e.* $h(\mathbf{Z}) = g(\mathbf{Z}, \mathbf{Z})$. Now, let the minimum of $g(\mathbf{X}, \mathbf{Z})$ over $\mathbf{X}$ be attained at $\mathbf{X}^\star$. The above assumptions imply the following chain of inequalities:

$$h(\mathbf{Z}) = g(\mathbf{Z}, \mathbf{Z}) \geq g(\mathbf{X}^\star, \mathbf{Z}) \geq h(\mathbf{X}^\star),$$

called the sandwich inequality. By substituting $\mathbf{Z}$ with $\mathbf{X}^\star$ and iterating this process, we obtain a sequence $(\mathbf{Z}_0, \ldots, \mathbf{Z}_t)$ that yields a non-increasing sequence of function values. (Borg & Groenen, 2005) showed that there is indeed such a majorizing function $g$ for the stress function, which is detailed in Section A of the Appendix. Through the lens of majorization, (De Leeuw, 1988) have shown that the above sequence converges to a local minimum of the stress minimization problem with a linear convergence rate. Local minima are more likely to occur in low-dimensional solutions while being less likely to happen in high dimensions (De Leeuw & Mair, 2009; Groenen & Heiser, 1996). The SMACOF algorithm based on stress majorization is summarized in Section A of the Appendix.

**Wasserstein Procrustes problem** For fixed embeddings $\mathbf{Z}$ and $\mathbf{Z}'$, the first two terms are independent of $\mathbf{P}$ and $\mathbf{O}$ and therefore this step consists in solving a Wasserstein Procrustes problem:

$$\min_{\mathbf{O}\in\mathcal{O}_d,\ \mathbf{P}\in\Pi(\mathbf{a},\mathbf{b})} \langle \mathbf{P}, d^2(\mathbf{Z}\mathbf{O}, \mathbf{Z}')\rangle_F - \varepsilon H(\mathbf{P}),$$

(8)

where we introduced an entropic regularization term which makes the objective strongly convex over $\mathbf{P}$ and thus easier to solve. The resulting problem can be solved with an alternating optimization

over $\mathbf{P}$ and $\mathbf{O}$ as studied in (Alvarez-Melis et al., 2019). Specifically, when fixing $\mathbf{O}$, the above problem is a classic discrete OT problem with entropic regularization that can be efficiently solved with, *e.g.*, the Sinkhorn's algorithm. More details about Sinkhorn's algorithm can be found in the Appendix. When fixing $\mathbf{P}$, the problem is equivalent to a classic orthogonal Procrustes problem: $\max_{\mathbf{O} \in \mathcal{O}_d} \langle \mathbf{O}, \mathbf{Z}^\top \mathbf{P} \mathbf{Z}' \rangle_F$, whose solution is simply given by the singular value decomposition of $\mathbf{Z}^\top \mathbf{P} \mathbf{Z}'$. Solving the Wasserstein Procrustes problem offers correspondences between two sets of embeddings of the same dimensions even if their spaces are not aligned, which is particularly suitable here. By alternating between these two steps, we obtain our final algorithm for solving Joint MDS.

**Overall algorithm** $\mathbf{Z}$ and $\mathbf{Z}'$ could be simply initialized with two individual SMACOF algorithms with $\mathbf{D}$ and $\mathbf{D}'$. However, for some hard matching problems, this might not offer a good initialization of the Wasserstein Procrustes problem. In this case, GW could be used as a convex relaxation of the Wasserstein Procrustes problem (Alvarez-Melis et al., 2019; Grave et al., 2019), which provides a better initialization for the coupling matrix $\mathbf{P}$. In addition, similar to the observations in (Alvarez-Melis et al., 2019), we also empirically find that annealing the entropic regularization parameter $\varepsilon$ in Wasserstein Procrustes leads to better convergence. When using the solution of GW as initialization, we also find that annealing $\lambda$ is useful. The full algorithm is summarized in Algorithm 1.

---

**Algorithm 1** Joint Multidimensional Scaling

---

1: **Input:** distances $\mathbf{D} \in \mathbb{R}^{n \times n}$ and $\mathbf{D}' \in \mathbb{R}^{n' \times n'}$, weights $\mathbf{W} \in \mathbb{R}^{n \times n}$ and $\mathbf{W}' \in \mathbb{R}^{n' \times n'}$, matching penalty $\lambda$, entropic regularization $\varepsilon$, max iterations $T$.
2: **Output:** low-dimensional embeddings $\mathbf{Z}, \mathbf{Z}' \in \mathbb{R}^{n \times d}$, optimal coupling $\mathbf{P} \in \mathbb{R}^{n \times n'}$.
3: Set $\mathbf{Z} = \text{MDS}(\mathbf{D}, \mathbf{W})$ and $\mathbf{Z}' = \text{MDS}(\mathbf{D}', \mathbf{W}')$ with SMACOF using random initialization.
4: Set $\tilde{\mathbf{D}}$ in Eq (7) with $\mathbf{D}$ and $\mathbf{D}'$.
5: **for** $t = 1, \ldots, T$ **do**
6:    Update $\mathbf{P}$ and $\mathbf{O}$ by solving Wasserstein Procrustes in Eq. (8) between $\mathbf{Z}$ and $\mathbf{Z}'$ using $\varepsilon$.
7:    Update $\mathbf{Z} = \mathbf{Z}\mathbf{O}$ and $\tilde{\mathbf{Z}}$ in Eq (7).
8:    Update $\tilde{\mathbf{W}}$ in Eq (7) using $\mathbf{P}$, $\lambda$ $\mathbf{W}$ and $\mathbf{W}'$.
9:    Update $\mathbf{Z}, \mathbf{Z}' = \text{MDS}(\tilde{\mathbf{D}}, \tilde{\mathbf{W}})$ using SMACOF with $\tilde{\mathbf{Z}}$ as initialization.
10: **end for**

---

**Complexity analysis** Let us denote by $N = n + n'$ the total number of data points of both domains. The complexity of a SMACOF iteration is the complexity of a Guttman Transform, which equals to $O(N^2)$ (De Leeuw, 1988). On the other hand, the complexity of one iteration of the alternating optimization method for solving the Wasserstein Procrustes is $O(d^3 + N^2)$ (Alvarez-Melis et al., 2019) where the first term corresponds to the complexity of SVD and the second term corresponds to the Sinkhorn's algorithm with a fixed number of iterations. While a linear convergence of the SMACOF algorithm was shown in (De Leeuw, 1988), we empirically observe that a fixed number of iterations is sufficient for the convergence of SMACOF as well as for solving Wasserstein Procrustes at each iteration of the Joint MDS algorithm 1. Therefore, for a total number of iterations $T$, the complexity of the full algorithm is $O(T(d^3 + 2N^2))$. Despite the quadratic complexity, our algorithm could benefit from the acceleration of GPUs as most of the operations in SMACOF and Sinkhorn's algorithm are matrix multiplications.

## 5 APPLICATIONS

### 5.1 JOINT VISUAL EXPLORATION OF TWO DATASETS

We show that our Joint MDS can be used to jointly visualize two related datasets by simultaneously mapping two datasets to a common 2D space while preserving the original manifold structures.

**Experimental setup** Here, we consider 3 (pairs of) synthetic datasets and 3 (pairs of) real-world datasets. The synthetic datasets respectively consist of a bifurcation, a Swiss roll and a circular frustum from (Liu et al., 2019). Each synthetic dataset has 300 samples that have been linearly projected onto 1000 and 2000 dimensional feature spaces respectively and have been injected white noise (Liu et al., 2019). The 3 real-world datasets consist of 2 sets of single-cell multi-omics data (SNAREseq and scGEM) (Demetci et al., 2022) and a subsampled digit dataset (MNIST-USPS (Deng, 2012; Hull, 1994)) . We compute the pairwise geodesic distances of each dataset as described in

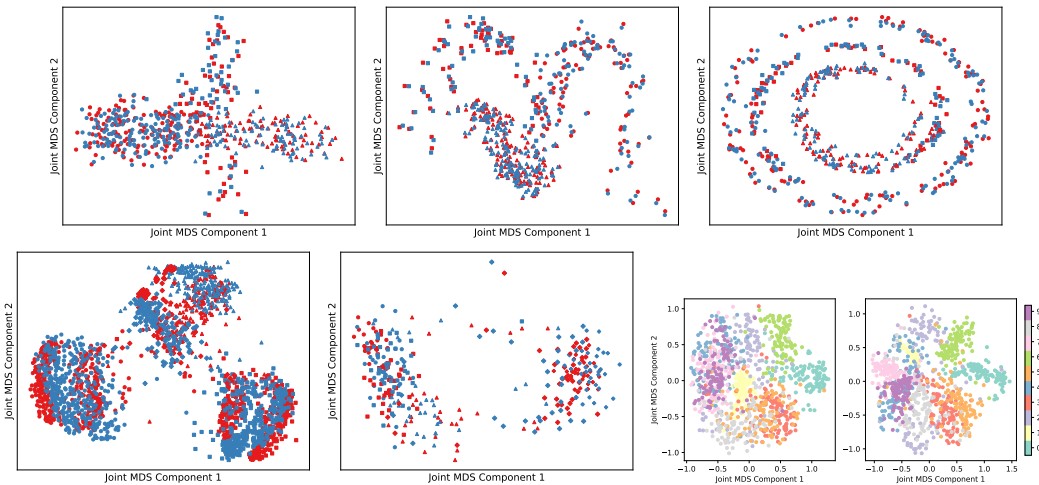

Figure 2: Joint visual exploration of two datasets. Top row: 3 synthetic datasets (bifurcation, Swiss roll, and circular frustum). Bottom row: real-world datasets (SNAREseq, scGEM and MNIST-USPS). Different colors represent datasets and different markers represent different classes except for MNIST-USPS, in which different colors represent different classes of digits from 0 to 9.

Section 4.2. We fix the number of components $d$ to 2 to visualize datasets in $\mathbb{R}^2$. We fix the matching penalization parameter $\lambda$ to 0.1 for all the datasets. More details about the datasets and parameters and additional results can be found in Section B.2 of the Appendix.

**Results** We show in Figure 2 the results for all the considered datasets. Joint MDS successfully embedded each pair of datasets to a common 2D space while preserving the original manifold structures such as proximity of similar classes. In contrast, some other recently proposed methods such as (Demetci et al., 2022) only map one dataset onto the space of the other dataset through a barycentric projection from GW, which could lose information about the manifold structure of the dataset before projection. As we also have access to the ground truth of the 1-to-1 correspondences for the synthetic and single-cell datasets, we can quantitatively assess the alignment learned by Joint MDS, by using the fraction of samples closer than the true match (FOSCTTM) metric (Liu et al., 2019). For a given sample from a dataset, we compute the fraction of samples from the other dataset that are closer to it than its true match and we average the fraction values for all the samples in both datasets. A lower average FOSCTTM generally indicates a better alignment and FOSCTTM would be zero for a perfect alignment as all the samples are closest to their true matches. We show in Table 1 the results of FOSCTTM compared to several baseline methods that are specifically proposed for aligning single-cell data. Joint MDS can also be used to align and visualize two sets of point clouds with a specific application for human pose alignment. We provide more details and results in Section B.2.5 of the Appendix due to limited space.

Table 1: Average FOSCTTM comparison (lower is better).

| Method | Bifurcation | Swiss roll | Circular frustum | SNAREseq | scGEM |
|---|---|---|---|---|---|
| MMD-MA (Liu et al., 2019) | 12.44 | 3.27 | 1.25 | 15.00 | 20.14 |
| UnionCom (Cao et al., 2020) | 8.30 | 1.57 | 15.20 | 26.50 | 20.96 |
| SCOT (Demetci et al., 2022) | 8.66 | 2.16 | 0.88 | **15.00** | 19.80 |
| Joint MDS (d=2) | 11.15 | 0.96 | 0.90 | 17.18 | 20.42 |
| Joint MDS (d=16) | **7.56** | **0.58** | **0.87** | 15.59 | **18.54** |

## 5.2 UNSUPERVISED HETEROGENEOUS DOMAIN ADAPTATION

We use the same datasets as in Section 5.1. To solve the HDA problem, we first solve the joint MDS problem between the two pairwise geodesic distance matrices, which provides embeddings for instances from both domains in the same space. Then, we train a $k$-NN classifier ($k$ is fixed to 5) on the source domain and estimate the labels on the target domain without any further adaptation. For the

more complex MNIST-USPS dataset, we use a linear SVM classifier with the regularization parameter set to 1 instead, which results in better prediction accuracy. We follow the same parameter selection as in (Demetci et al., 2022) and compare our method to the state-of-the-art unsupervised HDA methods including SCOT (Demetci et al., 2022) and EGW (Yan et al., 2018), which are both based on entropic regularized Gromov-Wasserstein. The classification accuracies are shown in Table 2. While all the methods work well on the easier datasets, our Joint MDS outperforms GW-based baselines on some harder tasks such as MNIST-USPS.

Table 2: Classification accuracy for unsupervised domain adaptation.

| Method | Bifurcation | Swiss roll | Circular frustum | SNAREseq | scGEM | MNIST-USPS |
|---|---|---|---|---|---|---|
| SCOT (Demetci et al., 2022) | 93.7 | 97.7 | **95.7** | **98.2** | 57.6 | 26.7 |
| EGW (Yan et al., 2018) | 95.7 | **99.3** | 94.7 | 93.8 | 62.7 | 43.1 |
| Joint MDS (d=2) | 96.0 | **99.3** | 94.0 | 85.5 | 64.4 | 15.0 |
| Joint MDS (d=16) | **96.7** | **99.3** | 94.7 | 94.7 | **72.9** | **60.2** |

## 5.3 GRAPH MATCHING

**Data** We apply the proposed method on two real-world datasets for the graph matching task which were used in (Xu et al., 2019b;a). The first dataset was collected from MIMIC-III (Johnson et al., 2016), which is one of the most commonly used critical care databases in healthcare related studies. This dataset contains more than 50,000 hospital admissions for patients and each is represented as a sequence of ICD (International Classification of Diseases) codes for different diseases and procedures. The interactions in the graph for disease (resp. procedure) are constructed with the diseases (resp. procedures) appearing in the same admission. The final two constructed graphs involve 11,086 admissions, one with 56 nodes of diseases and the other with 25 nodes of procedures. The procedure recommendation could be formulated as the problem of matching these two graphs. The second dataset contains the protein-protein interaction (PPI) networks of yeast. The original PPI network of yeast has 1,004 proteins together with 4,920 high-confidence interactions. The graph to match is its noisy version, which was built by adding $q\%$ lower-confidence interactions with $q \in \{5, 15, 25\}$.

**Experimental setup** To solve the graph matching problem, we use the shortest path distances on the graph as the input of our Joint MDS similarly to graph drawing methods (Gansner et al., 2004; Kamada et al., 1989), and use the coupling matrix $\mathbf{P}$ as the matching prediction. We compare our method with some recent graph matching methods, including MAGNA++ (Vijayan et al., 2015), HubAlign (Hashemifar & Xu, 2014), KerGM (Zhang et al., 2019) and GWL (Xu et al., 2019b). In particular, GWL is a powerful graph matching algorithm based on the Gromov-Wasserstein distance that outperforms many classic methods (Xu et al., 2019a; Liu et al., 2021). Note that Joint MDS differs from GWL in three aspects: i) Joint MDS relies on SMACOF which has convergence guarantees (De Leeuw, 1988), while GWL relies on SGD that does not necessarily converge for non-convex problems; ii) it generates isometric embeddings, which could be more beneficial for tasks require distance preservation such as protein structure alignment; iii) it uses OT instead of GW in the low-dimensional embedding space to learn the correspondences. For baseline methods, we use the authors' implementations with recommended hyperparameters. For our method, we follow the same hyperparameter selection procedure and report the average accuracy over 5 different runs. We do not perform MAGNA++ and HubAlign on MIMIC-III matching task since they can only produce one-to-one node pairs.

For the diseases-procedure graph matching task, we use 75% of admissions for training and 25% for testing. We use training data to respectively build disease graph and procedure graph, and use the test data to build the correspondences between disease and procedure as the true recommendation. For each node in the disease graph, we calculate top 3 and top 5 recommended procedure nodes based on the coupling matrix $\mathbf{P}$. We consider it as a correct match if the true closest procedure node appears in the predicted top-k recommendation list. For the PPI network matching task, we calculated the node correctness (NC) as the evaluation metrics: $\sum_{i=1}^{n} \sum_{j=1}^{n} P_{ij}$ if $T_{ij} = 1$, where $P$ and $T$ respectively corresponds to the predicted matching probability matrix and the true matching matrix.

**Results** In Table 3, the results of the two graph matching tasks are presented. Our method achieves comparable performance to GWL, while both methods outperform other classic baselines. More results and experimental details are available in Section B.4 of the Appendix.

Table 3: Graph matching performances on PPI networks and MIMIC-III disease-procedure graphs.

| Method | PPI 5% | PPI 15% | PPI 25% | MIMIC top 3 | MIMIC top 5 |
|---|---|---|---|---|---|
| MAGNA++ | 50.00 | 35.16 | 12.85 | — | — |
| HubAlign | 46.06 | 32.47 | 27.39 | — | — |
| KerGM | 66.14 | 39.04 | 32.17 | 22.67 | **47.86** |
| GWL | 84.31 | **74.35** | **67.42** | 27.98 | 42.14 |
| Joint MDS | **86.44±0.33** | 72.31±0.62 | 55.3±0.78 | **30.24±1.66** | 46.28±1.51 |

| Method | RMSD-D↓ |
|---|---|
| GUMA-2D | 28.89 |
| GUMA | 26.04 |
| Joint MDS-2D | 18.04±0.10 |
| Joint MDS | 13.11±0.02 |

Figure 3: Protein structure alignment results. Left: average RMSD-D over 58 pairs of protein models. Right: RMSD-D comparison of GUMA (Cui et al., 2014) and Joint MDS, with the difference significance measured by a Wilcoxon signed-rank test.

## 5.4 PROTEIN STRUCTURE ALIGNMENT

Protein 3D structure alignment is an important sub-task in protein structure reconstruction (Wang & Mahadevan, 2009; Badaczewska-Dawid et al., 2020) and protein model quality assessment (Pereira et al., 2021). This task consists of structurally aligning two moderately accurate protein models estimated by different methods, either experimental or computational, and eventually integrating them to generate a more accurate and high-resolution 3D model of protein folding. Here, we consider a more challenging setting where the correspondences across protein models are not known. In contrast to previous work (Wang & Mahadevan, 2009; Cui et al., 2014) where only few examples are illustrated, we use a larger and more realistic dataset, and provide a quantitative way to evaluate the unsupervised manifold alignment methods for this task. Specifically, we consider here the first domain of all the proteins from CASP14 (Pereira et al., 2021) from T1024 to T1113 and use the protein models predicted by the two top-placed methods in the leaderboard, namely AlphaFold2 and Baker. This results in a dataset of 58 pairs of protein models, with an average number of residues equal to 198. More details about the dataset can be found in Section B.5 of the Appendix. We compare our method to a state-of-the-art unsupervised manifold alignment method namely GUMA (Cui et al., 2014). The parameters are fixed across the dataset and the performance is measured as the average of the RMSD-D obtained by 3 different runs, defined as the root-mean-square deviation (RMSD-D) of residue positions and distances, given by

$$\text{RMSD-D} = \sqrt{\frac{1}{n^2}\|\mathbf{D} - d(\mathbf{Z},\mathbf{Z})\|_F^2} + \sqrt{\frac{1}{n^2}\|\mathbf{D}' - d(\mathbf{Z}',\mathbf{Z}')\|_F^2} + \sqrt{\frac{1}{n}\sum_{i,j=1}^n \mathbf{P}_{ij}\|\mathbf{z}_i - \mathbf{z}_j'\|^2},$$

where $\mathbf{P}$ denotes the true permutation matrix with 0 and 1 as entries. We respectively compute embeddings in the orignal 3D space and in the 2D space, and show the results in Figure 3. We also measure the significance of the difference with the $p$-value of a Wilcoxon signed-rank test. While GUMA hardly works in this challenging task with few number of RMSD-D values being small, our Joint MDS manages to produce good alignment in a large number of proteins.

## 6 CONCLUSION

We introduced Joint MDS, a flexible and powerful analysis tool for joint exploration of two datasets from different domains without known correspondences between instances across the datasets. While it only requires pairwise intra-dataset dissimilarities as input, we showcased its applicability and effectiveness in several applications. We see the main limitation of our approach in its quadratic complexity of number of samples, which makes it hard to scale to very large datasets. Investigation of faster stochastic and online optimization methods for solving either MDS or Wasserstein Procrustes (Rajawat & Kumar, 2017; Pai et al., 2019; Shamai et al., 2015; Boyarski et al., 2017) would be interesting research directions.

ACKNOWLEDGEMENT

This project has received funding from the European Union's Horizon 2020 research and innovation programme under the Marie Sklodowska-Curie grant agreement No 813533 (K.B.). The authors would also like to thank Leslie O'Bray for her insightful feedback on the manuscript, which greatly improved it.

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

# Appendix

This appendix provides additional background and details about methods and experiments. It is organized as follows: Section A provides further background on optimization and Section B presents experimental details and additional results.

## A   FURTHER BACKGROUND ON OPTIMIZATION

### A.1   MULTIDIMENSIONAL SCALING AND SMACOF

This section provides further background on the metric MDS problem and details about the SMACOF algorithm. The metric MDS problem consists in solving the minimization problem of the stress function defined in Eq. (1), or equivalently

$$\min_{\mathbf{Z} \in \mathbb{R}^{n \times d}} \text{stress}(\mathbf{Z}, \mathbf{D}, \mathbf{W}) := \sum_{i<j}^{n} w_{ij}(d_{ij} - d(\mathbf{z}_i, \mathbf{z}_j))^2, \tag{9}$$

for symmetric dissimilarities. We first give the explicit expression of the majorizing function for the above stress function.

**Lemma 1** ((Borg & Groenen, 2005)). *Let us define*

$$g(\mathbf{X}, \mathbf{Z}) := \sum_{i<j} w_{ij}d_{ij}^2 + \sum_{i<j} w_{ij}d^2(\mathbf{x}_i, \mathbf{x}_j) - 2\sum_{i<j} w_{ij}d_{ij} \frac{\langle \mathbf{x}_i - \mathbf{x}_j, \mathbf{z}_i - \mathbf{z}_j \rangle}{d(\mathbf{z}_i, \mathbf{z}_j)}.$$

*Then we have* $\text{stress}(\mathbf{X}, \mathbf{D}, \mathbf{W}) \leq g(\mathbf{X}, \mathbf{Z})$ *and* $\text{stress}(\mathbf{Z}, \mathbf{D}, \mathbf{W}) = g(\mathbf{Z}, \mathbf{Z})$.

*Proof.* First, we have by Cauchy-Schwarz inequality:

$$\langle \mathbf{x}_i - \mathbf{x}_j, \mathbf{z}_i - \mathbf{z}_j \rangle \leq \|\mathbf{x}_i - \mathbf{x}_j\|\|\mathbf{z}_i - \mathbf{z}_j\| = d(\mathbf{x}_i, \mathbf{x}_j)d(\mathbf{z}_i, \mathbf{z}_j).$$

By expanding the expression of the stress and using the above inequality, we obtain

$$\begin{aligned}
\text{stress}(\mathbf{X}, \mathbf{D}, \mathbf{W}) &= \sum_{i<j} w_{ij}d_{ij}^2 + \sum_{i<j} w_{ij}d^2(\mathbf{x}_i, \mathbf{x}_j) - 2\sum_{i<j} w_{ij}d_{ij}d(\mathbf{x}_i, \mathbf{x}_j) \\
&\leq \sum_{i<j} w_{ij}d_{ij}^2 + \sum_{i<j} w_{ij}d^2(\mathbf{x}_i, \mathbf{x}_j) - 2\sum_{i<j} w_{ij}d_{ij} \frac{\langle \mathbf{x}_i - \mathbf{x}_j, \mathbf{z}_i - \mathbf{z}_j \rangle}{d(\mathbf{z}_i, \mathbf{z}_j)} \\
&= g(\mathbf{X}, \mathbf{Z}).
\end{aligned}$$

It is also easy to show that when $\mathbf{X} = \mathbf{Z}$, we have equality. $\square$

Now the following properties of $g$ provides simple update at each iteration of the stress majorization:

**Lemma 2** ((Borg & Groenen, 2005)). *$g(\mathbf{X}, \mathbf{Z})$ is a quadratic function in $\mathbf{X}$ whose minimum is attained through the Guttman transform at $\mathbf{X}^\star = \mathbf{V}^+\mathbf{B}(\mathbf{Z})\mathbf{Z}$, where $\mathbf{V} = \sum_{i<j} w_{ij}(\mathbf{e}_i - \mathbf{e}_j)(\mathbf{e}_i - \mathbf{e}_j)^\top$ with $\mathbf{e}$ being the canonical basis of $\mathbb{R}^n$, $\mathbf{V}^+$ denoting its pseudo-inverse, and $\mathbf{B}(\mathbf{Z}) = \sum_{i<j} b_{ij}(\mathbf{e}_i - \mathbf{e}_j)(\mathbf{e}_i - \mathbf{e}_j)^\top$ with*

$$b_{ij} = \begin{cases} w_{ij}d_{ij}/d(\mathbf{z}_i, \mathbf{z}_j) & \text{if } d(\mathbf{z}_i, \mathbf{z}_j) > 0, \\ 0 & \text{otherwise.} \end{cases}$$

*Proof.* Through simple calculus, we have

$$\sum_{i<j} w_{ij}d^2(\mathbf{x}_i, \mathbf{x}_j) = \sum_{i<j} w_{ij}\|\mathbf{x}_i - \mathbf{x}_j\|^2 = \text{tr}(\mathbf{X}^\top \mathbf{V}\mathbf{X}),$$

and

$$\sum_{i<j} w_{ij} d_{ij} \frac{\langle \mathbf{x}_i - \mathbf{x}_j, \mathbf{z}_i - \mathbf{z}_j \rangle}{d(\mathbf{z}_i, \mathbf{z}_j)} = \mathrm{tr}(\mathbf{X}^\top \mathbf{B}(\mathbf{Z})\mathbf{Z}).$$

Since $g$ is a quadratic function in $\mathbf{X}$, its minimum is simply attained at its stationary point, equal to

$$\mathbf{X}^\star = \mathbf{V}^+ \mathbf{B}(\mathbf{Z})\mathbf{Z}.$$

$\square$

Based on these two Lemmas and the principle of the majorization, we can obtain the SMACOF algorithm for solving the metric MDS problem, summarized in Algorithm 2.

---

**Algorithm 2** SMACOF algorithm

1: **Input:** distance $\mathbf{D} \in \mathbb{R}^{n \times n}$, weight $\mathbf{W} \in \mathbb{R}^{n \times n}$, initial embeddings $\mathbf{Z}_0$, tolerance $\varepsilon$, max iterations $T$.
2: **Output:** low-dimensional embeddings $\mathbf{Z} \in \mathbb{R}^{n \times d}$.
3: **for** $t = 1, \ldots, T$ **do**
4:     Compute the Guttman transform $\mathbf{Z}_t = \mathbf{V}^+ \mathbf{B}(\mathbf{Z}_{t-1})\mathbf{Z}_{t-1}$.
5:     **if** $\mathrm{stress}(\mathbf{Z}_{t-1}, \mathbf{D}, \mathbf{W}) - \mathrm{stress}(\mathbf{Z}_t, \mathbf{D}, \mathbf{W}) < \varepsilon$ **then**
6:         Set $\mathbf{Z} = \mathbf{Z}_t$.
7:         **break**
8:     **end if**
9: **end for**

---

### A.2 WASSERSTEIN PROCRUSTES

The section provides further background and details about the Wasserstein Procrustes problem (Alvarez-Melis et al., 2019).

#### A.2.1 SINKHORN'S ALGORITHM

When fixing $\mathbf{O}$ in Eq. (8), the problem amounts to solving the classic discrete OT problem with entropic regularization:

$$\min_{\mathbf{P} \in \Pi(\mathbf{a},\mathbf{b})} \langle \mathbf{P}, \mathbf{C} \rangle_F - \varepsilon H(\mathbf{P}), \tag{10}$$

where $H(\mathbf{P}) := -\sum_{ij} \mathbf{P}_{ij}(\log \mathbf{P}_{ij} - 1)$ and $\mathbf{C} = d^2(\mathbf{Z}\mathbf{O}, \mathbf{Z}')$ is the cost matrix. This problem can be efficiently solved by the Sinkhorn's algorithm, which is an iterative matrix scaling method to approach the double stochastic matrix. Specifically, the $\ell$-th iteration of the algorithm performs the following updates:

$$\mathbf{u}^{(\ell)} = \frac{\mathbf{a}}{\mathbf{K}\mathbf{v}^{(\ell)}}, \qquad \mathbf{v}^{(\ell+1)} = \frac{\mathbf{b}}{\mathbf{K}^\top \mathbf{u}^{(\ell)}},$$

where $\mathbf{K} = e^{-\mathbf{C}/\varepsilon}$. Then, the matrix $\mathrm{diag}(\mathbf{u}^{(\ell)})\mathbf{K}\,\mathrm{diag}(\mathbf{v}^{(\ell)})$ converges to the solution of Eq. (10) when $\ell$ tends to $\infty$. This algorithm converges faster with larger $\varepsilon$ as strong regularization leads to a more convex objective (Peyré et al., 2019). As a consequence, using an annealing scheme on $\varepsilon$ has been observed useful for alleviating the initialization issue when jointly optimizing on $\mathbf{P}$ and $\mathbf{O}$ in (Alvarez-Melis et al., 2019). Thus, we also adopt the same annealing scheme $\varepsilon_t = \alpha \varepsilon_{t-1}$ in our experiments with a decay factor $\alpha = 0.95$.

#### A.2.2 ORTHOGONAL PROCRUSTES

When fixing $\mathbf{P}$, the problem in Eq. (8) is reduced to

$$\max_{\mathbf{O} \in \mathcal{O}_d} \langle \mathbf{O}, \mathbf{M} \rangle_F,$$

where $\mathbf{M} = \mathbf{Z}^\top \mathbf{P}\mathbf{Z}'$. The solution is given by the following Lemma

**Lemma 3** ((Alvarez-Melis et al., 2019)). *Let $\mathbf{M}$ be the matrix with SVD $\mathbf{M} = \mathbf{U}\boldsymbol{\Lambda}\mathbf{V}^\top$, then* $\arg\max_{\mathbf{O} \in \mathcal{O}_d} \langle \mathbf{O}, \mathbf{M} \rangle_F = \mathbf{U}\mathbf{V}^\top$.

*Proof.* The proof is simply based on the fact that the set of orthogonal matrices is a group. Specifically, we have

$$\langle \mathbf{O}, \mathbf{M} \rangle_F = \langle \mathbf{O}, \mathbf{U}\boldsymbol{\Lambda}\mathbf{V}^\top \rangle_F = \langle \mathbf{U}^\top \mathbf{O}\mathbf{V}, \boldsymbol{\Lambda} \rangle_F,$$

which is maximized when $\mathbf{U}^\top \mathbf{O}\mathbf{V}$ is equal to the identity matrix, since it is an orthogonal matrix (as the product of orthogonal matrices). As a consequence, $\mathbf{O} = \mathbf{U}\mathbf{V}^\top$. □

# B    EXPERIMENTAL DETAILS AND ADDITIONAL RESULTS

In this section, we provide implementation details and additional experimental results. Our code will be released upon publication.

## B.1    COMPUTATION DETAILS

All experiments are run on a single Macbook Pro 2020 laptop with a 2 GHz Quad-core Intel core i5 CPU.

## B.2    JOINT VISUAL EXPLORATION OF TWO DATASETS

We provide details about datasets, implementation and evaluation metric for joint visual exploration of two datasets.

### B.2.1    DATASETS

The datasets used in joint visual exploration include 3 (pairs of) synthetic datasets and 3 (pairs of) real-world datasets.

**Synthetic datasets.**    We use 3 synthetic datasets from (Liu et al., 2019)[1], including a bifurcation, a Swiss roll, and a circular frustum. All 3 synthetic datasets consist of two domains of 300 samples that have been linearly projected onto 1000 and 2000-dimensional feature spaces respectively where the mapping matrix is sampled from a Gaussian distribution. The transformed samples are also injected small white noise (Liu et al., 2019). There are respectively 3 classes for each dataset. Each resulting dataset thus provides two data matrices $\mathbf{X} \in \mathbb{R}^{300 \times 1000}$ and $\mathbf{X}' \in \mathbb{R}^{300 \times 2000}$ associated with their respective class labels $\mathbf{y}$ and $\mathbf{y}'$ which will be used for the unsupervised heterogeneous domain adaptation (HDA) problem. All 3 datasets were simulated with one-to-one sample-wise correspondences, which are solely used for evaluation purpose. Each domain is projected to a different dimension, so there is no feature-wise correspondence either. In all simulations, we normalize the features with standardisation before running the Joint MDS as in (Liu et al., 2019).

**Real-world datasets.**    We use 2 sets of single-cell multi-omics data from (Demetci et al., 2022) and a digit dataset to demonstrate the applicability of our method to real datasets.

The 2 single-cell datasets are generated by co-assays, which provides cell-level correspondence information for evaluation. The first dataset is generated by the scGEM assay, which simultaneously profiles gene expression and DNA methylation. It results in two data matrices $\mathbf{X} \in \mathbb{R}^{177 \times 34}$ for the gene expression data and $\mathbf{X} \in \mathbb{R}^{177 \times 27}$ for chromatin accessibility data associated with 5 classes based on the cell types: iPS, d24T+, d16T+, d8 and BJ. The other dataset is generated by the SNAREseq assay, linking gene expression with chromatin accessibility. The data has been pre-processed in (Demetci et al., 2022) and results in two data matrices $\mathbf{X} \in \mathbb{R}^{1047 \times 19}$ and $\mathbf{X}' \in \mathbb{R}^{1047 \times 10}$ respectively for ATAC-seq and RNA-seq, associated with 4 classes based on the cell types: GM, BJ, K562 and H1. Both datasets are unit normalized as in (Demetci et al., 2022).

The digit dataset MNIST-USPS contain two subsets of MNIST and USPS respectively. Each subset include 1000 digit images with 100 images for each digit class, which results in two data matrices $\mathbf{X} \in \mathbb{R}^{1000 \times 784}$ and $\mathbf{X}' \in \mathbb{R}^{1000 \times 256}$ for subsets of MNIST and USPS respectively. There are 10 classes in total representing digits from 0 to 9. The data is not preprocessed but only raw pixels are used to compute the distance matrices.

---

[1]The datasets can be downloaded from https://noble.gs.washington.edu/proj/mmd-ma/

### B.2.2 IMPLEMENTATION DETAILS

To compute the dissimilarities for data from each domain, we use the geodesic distances presented in Section 4.2. The resulting distance matrices is rescaled by the average of the shortest-path distances to match the distances scale of the two domains. The number of nearest neighbors (NN) for each dataset is set to minimize the FOSCTTM via a grid search similar to (Demetci et al., 2022), and is then fixed throughout the experiments. For MNIST-USPS, since we do not have 1-to-1 sample-wise correspondences, we fix number of NNs to 5. For synthetic datasets, the matching penalization parameter $\lambda$ is fixed to 0.1 and the entropic regularization parameter $\varepsilon$ for OT is fixed to 1.0 with an annealing decay equal to 0.95. For real-world datasets, we select both parameters based on either FOSCTTM if 1-to-1 correspondences are available or transfer accuracy otherwise. We run Joint MDS 4 times with different initializations and select the best embeddings determined by the run with the smallest final objective value to mitigate the local minima issues. We found our results generally stable and thus did not report the std for these datasets.

### B.2.3 EVALUATION METRIC

As we also have access to the ground truth of the 1-to-1 correspondences for the 3 synthetic datasets and the 2 single-cell datasets, we can quantitatively assess the alignment learned by Joint MDS. This is achieved by using the fraction of samples closer than the true match (FOSCTTM) metric (Liu et al., 2019). For a given sample from a dataset, we compute the fraction of samples from the other dataset that are closer to it than its true match and we average the fraction values for all the samples in both datasets. A lower average FOSCTTM generally indicates a better alignment and FOSCTTM would be zero for a perfect alignment as all the samples are closest to their true matches.

### B.2.4 ADDITIONAL VISUAL RESULTS

We show in Figure 4 visualization comparisons of MDS and Joint MDS on the 6 datasets. The MDS visualizations are obtained by running the SMACOF algorithm individually on each domain. While there's no alignment between the samples across the domains, our Joint MDS clearly aligns the samples in the 2D space as data points from the same class are located at similar positions across the two domains.

### B.2.5 HUMAN BODY POSE ALIGNMENT

In addition to exploring two datasets, Joint MDS can also be used for comparing two sets of point clouds. In order to make our discussion visually more relatable, we apply Joint MDS to the task of aligning human body point clouds in two different poses. We consider here 3 pairs of human body poses, 2 pairs from the MPI FAUST dataset (Bogo et al., 2014) (each pose has 3446 points) and 1 pair from Solomon et al. (2016) (has 1024 and 1502 points respectively) to visualize how Joint MDS works in different cases. For the sake of simplicity, we only used Euclidean distances as the pairwise dissimilarities. We fixed $\lambda$ to 0.1 and entropic regularization $\varepsilon$ to 1.0 with annealing. We respectively compute embeddings in 3D and 2D space, and visualize the correspondences found by Joint MDS. The results are shown in Figure 5 where each column represents an example. Different colors indicate a different position $i$ in the source set and the alpha value of the corresponding color in the target set is given by the correspondence vector $\mathbf{P}_i$ returned by Joint MDS. Joint MDS is able to find correspondences between two poses, even if the target set of point clouds only shares semantic structure as shown in the right column.

### B.3 UNSUPERVISED HETEROGENEOUS DOMAIN ADAPTATION

We use the same datasets as in Section B.2 as all the datasets also provide class labels. We use the same hyperparameters as in Section B.2, including the number of NNs for computing the geodesic distance matrices, the matching penalization parameter and the entropic regularization parameter. To handle the classification problem, we first use our Joint MDS to obtain two embeddings for both domains. Then, we train a simple $k$-NN classifier (with $k$ fixed to 5) on the embeddings of domain 1 and evaluate the classifier on domain 2 without any further adaptation. For the more complex dataset MNIST-USPS, we use a linear SVM classifier with the regularization parameter $C = 1$, which we found substantially outperforming the $k$-NN classifier. In addition, we also observe that the transfer

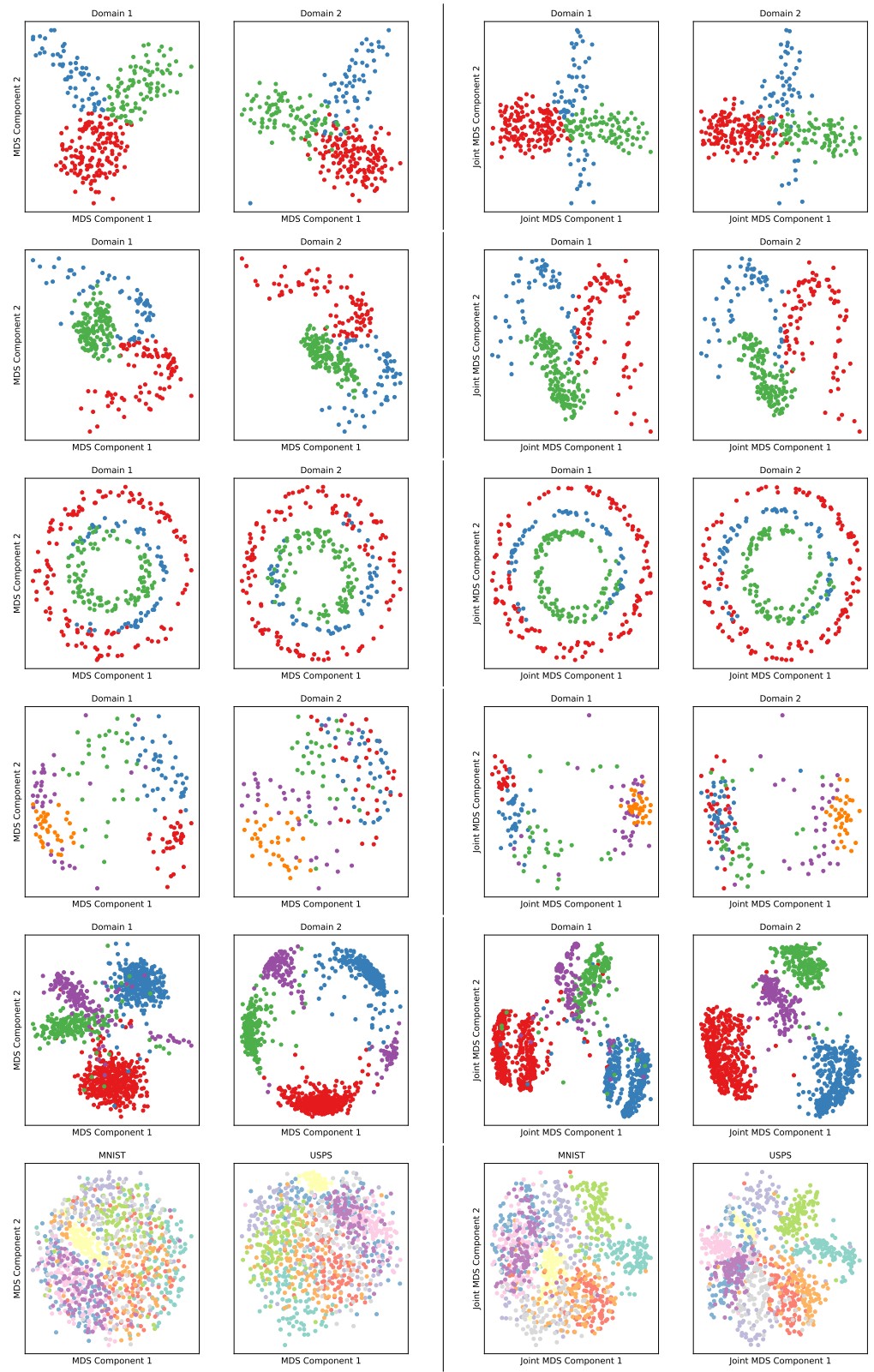

Figure 4: Comparison of MDS and Joint MDS. Each row presents a dataset and from top to bottom the datasets are respectively: bifurcation, Swiss roll, circular frustum, scGEM, SNAREseq and MNIST-USPS. Left column: results obtained by MDS applied individually to each domain with the Euclidean distances. Right column: results obtained by Joint MDS. Each color represents a class.

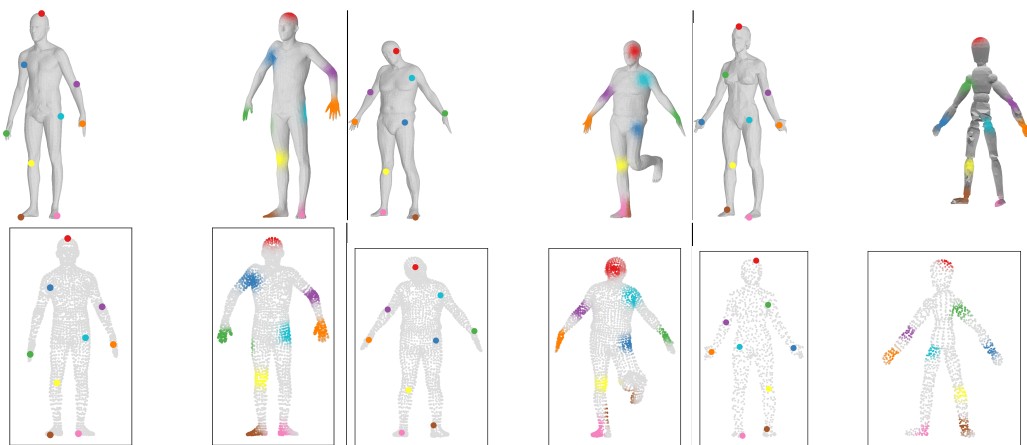

Figure 5: Joint MDS can find correspondences between a source set of point clouds and a target set of point clouds with a similar structure or with a shared semantic structure (right column), as well as a common embedding space. Note that here we plot the mesh surfaces in the 3D embeddings for visual purpose and we did not use any mesh information for correspondence finding but only point clouds.

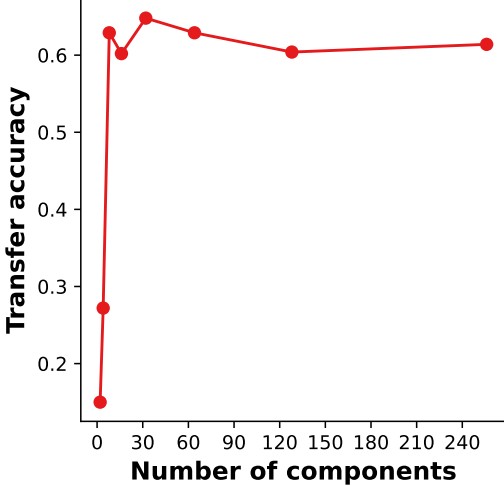

Figure 6: Transfer accuracy vs number of components for Joint MDS on the MNIST-USPS dataset.

accuracy generally increases with the number of components $d$, as shown on MNIST-USPS as an example in Figure 6.

## B.4 GRAPH MATCHING

Here, we provide the dataset and implementation details for the experiments with respect to the graph matching task.

### B.4.1 DATASETS AND BASELINES

We use two graph matching datasets including a PPI network of yeast (with its noisy versions) and the MIMIC-III disease-procedure interaction dataset. The two datasets can be downloaded at https://github.com/HongtengXu/s-gwl and https://github.com/HongtengXu/gwl respectively.

For each baseline method to compare with, we list its source and language below:

- HubAlign (C) (Hashemifar & Xu, 2014): `https://home.ttic.edu/~hashemifar/`
- MAGNA++ (C) (Vijayan et al., 2015): `https://www3.nd.edu/~cone/MAGNA++/`
- GWL (Python) (Xu et al., 2019b): `https://github.com/HongtengXu/s-gwl`

All baselines are implemented under the recommended settings.

### B.4.2 Evaluation metrics

As described in section 5.3, the PPI network matching task is a one-to-one node matching task, where we can directly compute the node correctness (NC) based on the true matching matrix and predicted matching matrix. However, the number of diseases (56) and the number of procedures (25) do not match in the disease-procedure graph matching task. In contrast to the original implementation in (Xu et al., 2019b) by Xu et al., which evaluates the learnt matching matrix by comparing the recommended procedures with the true procedures for each individual admission from the test set, we propose to build the disease and procedure graphs from these individuals and evaluate the recommended procedures for each disease directly. We compare the top 3 and top 5 recommended procedures with the corresponding true procedures, which has the most interactions with the target disease in admissions from the test set, then we take the average accuracy for all disease nodes as the evaluation metric.

### B.4.3 Implementation details

We first normalize the adjacency matrices for each graph via $\mathbf{D}^{-1/2}\mathbf{A}\mathbf{D}^{-1/2}$ where $\mathbf{A}$ denotes the adjacency matrix and $\mathbf{D}$ denotes the diagonal matrix of degrees. Then, we compute the shortest-path distances as the dissimilarities as described in Section B.2.2. Similar to stress majorization-based graph drawing algorithms (Gansner et al., 2004; Kamada et al., 1989), we apply different weights $\mathbf{W} \in \mathbb{R}^{n \times n}$ and $\mathbf{W}' \in \mathbb{R}^{n' \times n'}$ on the two distance deviation terms in the weighted stress minimization problem (5) with $w_{ij} = 1/d_{ij}^k$ and $d_{ij}$ is the shortest-path distance between node $i$ and node $j$ (similarity, for $w'_{ij} = 1/d_{ij}'^k$). Empirically, we find $k = 4$ leads to best performances.

### B.4.4 Rationality analysis

To further verify the rationality of the learnt matching matrix of the proposed method, we visualize the $k$-NN graph of disease nodes and procedure nodes in the common low-dimensional space in Figure 7 and we can further interpret the learned embeddings from a clinical point of view. For example, in the selected region inside the red dash line box, we can see a cluster of cardiac-related diseases like congestive heart failure (d4280), pleural effusion (d5119), Percutaneous transluminal coronary angioplasty (dV4582), and also cardiac-related procedures like extracorporeal circulation auxiliary to open heart surgery (p3961), combined right and left heart cardiac catheterization (p3723) and single internal mammary-coronary artery bypass (p3615). For reference, the complete ICD codes for the diseases and procedures can be found in the supplementary material of (Xu et al., 2019b). This finding indicates that our method can learn clinically meaningful joint embeddings from the disease and procedure graphs.

### B.5 Protein structure alignment

Protein structure alignment takes two protein 3D structure models as input, and outputs two new protein structure models that preserve the pairwise distances while being aligned.

**Datasets and experimental setup.** In contrast to the previous work (Wang & Mahadevan, 2009; Cui et al., 2014) where only one protein was studied, we use a much larger and more realistic dataset, and provide a quantitative way to evaluate the unsupervised manifold alignment methods for this task. Specifically, we consider here the first domain of all the proteins from CASP14 (Pereira et al., 2021) from T1024 to T1113 and use the protein models predicted by the two top-placed methods in the leaderboard, namely AlphaFold2 and Baker. This results in a dataset of 58 pairs of protein models, with an average number of residues equal to 198. The dataset can be downloaded from `https://predictioncenter.org/casp14/results.cgi`.

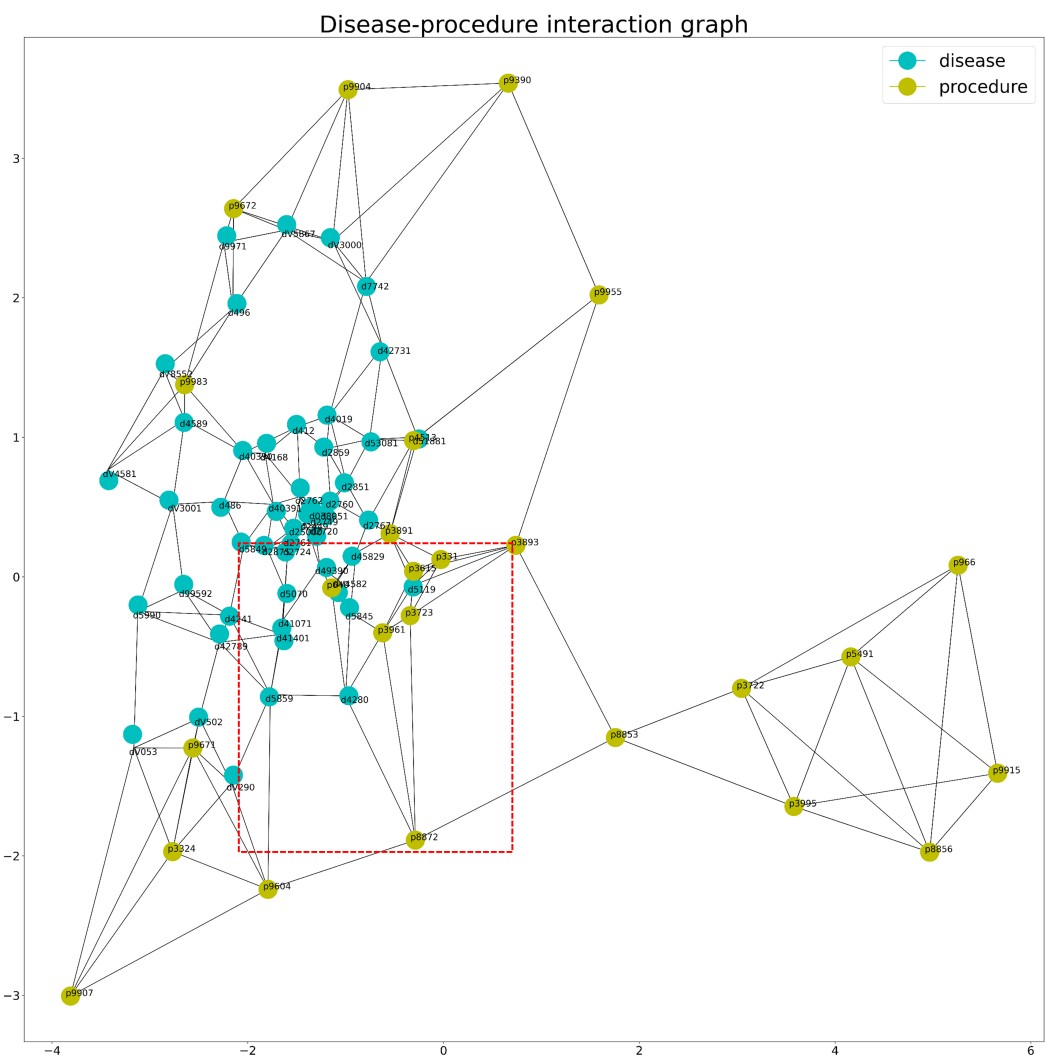

Figure 7: Visualization of the $k$-NN graph of diseases and procedures in the common 2D space with $k = 4$.

Table 4: Performance comparison on remove the orthogonal Procrustes matrix

| Method | scGEM | SNAREseq | PPI 5% | MIMIC top 3 | MIMIC top 5 |
|--------|-------|----------|--------|-------------|-------------|
| Removing $\mathbf{O}$ | 61.8 | 94.1 | 85.25 | 14.48 | 35.48 |
| Original | 72.9 | 94.7 | 86.44 | 30.24 | 46.28 |

We compare Joint MDS to the state-of-the-art unsupervised manifold alignment method GUMA (Cui et al., 2014). We use Joint MDS directly on the pairwise Euclidean distances between the residue 3D coordinates. The parameters for both our method and GUMA are fixed throughout the datasets, including $\lambda$ fixed to 0.1, entropic regularization $\varepsilon$ fixed to 10. We use the Gromov-Wasserstein initialization as described in Section 4.3. As MDS could be affected by bad initialization, we run Joint MDS 3 times on each protein pair with different random seeds and take the embeddings with the smallest cost for evaluation. To measure the robustness of our method, we repeat this process 3 times again, and take the average and standard deviation of the RMSD-D across all repetitions. The small standard deviation in Figure 3 demonstrate the robustness of our method.

**Additional alignment results.**   We showcase in Figure 8 five examples of alignment (TS1024, TS1039, TS1050, TS1058 and TS1099) obtained by GUMA, Joint MDS, GUMA-2D and Joint MDS-2D. While Joint MDS outperforms GUMA in almost all datasets, we also find a few protein samples where both methods fail.

## B.6   ADDITIONAL EXPERIMENTS

**Runtime comparison.**   We use four datasets to compare the runtime of Joint MDS with other baseline methods. The four datasets include: scGEM and SNAREseq from (Demetci et al., 2022) for unsupervised heterogeneous domain adaptation, MIMIC (Xu et al., 2019b) and PPI network (5% noise) (Xu et al., 2019a) for graph matching. The baseline methods we compared against are SCOT (Demetci et al., 2022), UnionCom (Cao et al., 2020), EGW (Yan et al., 2018) (for unsupervised heterogeneous domain adaptation) and GWL (Xu et al., 2019b) (for graph matching). For all baseline methods, we used the original implementations for each task. For our method, we used the same parameters which achieved the reported performance. We ran all the methods until convergence and repeated this three times to compare the average consumed time. Figure 9 suggest that the runtime of Joint MDS is at the same order of magnitude compared to the best of other baseline methods while achieving the best performance in most of the tasks.

**Convergence curves.**   Here, we provide the convergence curves (objective value versus number of iterations) of our method on the above four tasks. The curves in figure 10 show that the proposed optimization algorithm reached final convergence for all the tasks. Empirically, we also validated the convergence of the method by visualizing the node correctness of each iteration in the PPI network matching task for the three noise levels respectively, in Figure 11. Even though the coupling matrix $\mathbf{P}$ is intialized with GW which already offers a not bad quality, its quality further improves with the number of iterations of Joint MDS, thanks to regularization through explicit learning of embeddings.

**Removing the orthogonal Procrustes matrix.**   We further evaluate the necessity of adding the orthogonal Procrustes matrix $\mathbf{O}$ in our final objective function, as showed in Eq 5, by an ablation study comparing the performance of removing $\mathbf{O}$ and the performance of the original objective. Similarly to the runtime analysis, the ablation study was conducted on two unsupervised heterogeneous domain adaptation tasks and two graph matching tasks. In Table 4 we can see that the original Joint MDS has achieved consistent higher performance in various tasks compared its incomplete version with $\mathbf{O}$ removed, on some tasks like scGEM and MIMIC, the performance gap is very large, which demonstrates the importance of incorporating $\mathbf{O}$ in the objective function.

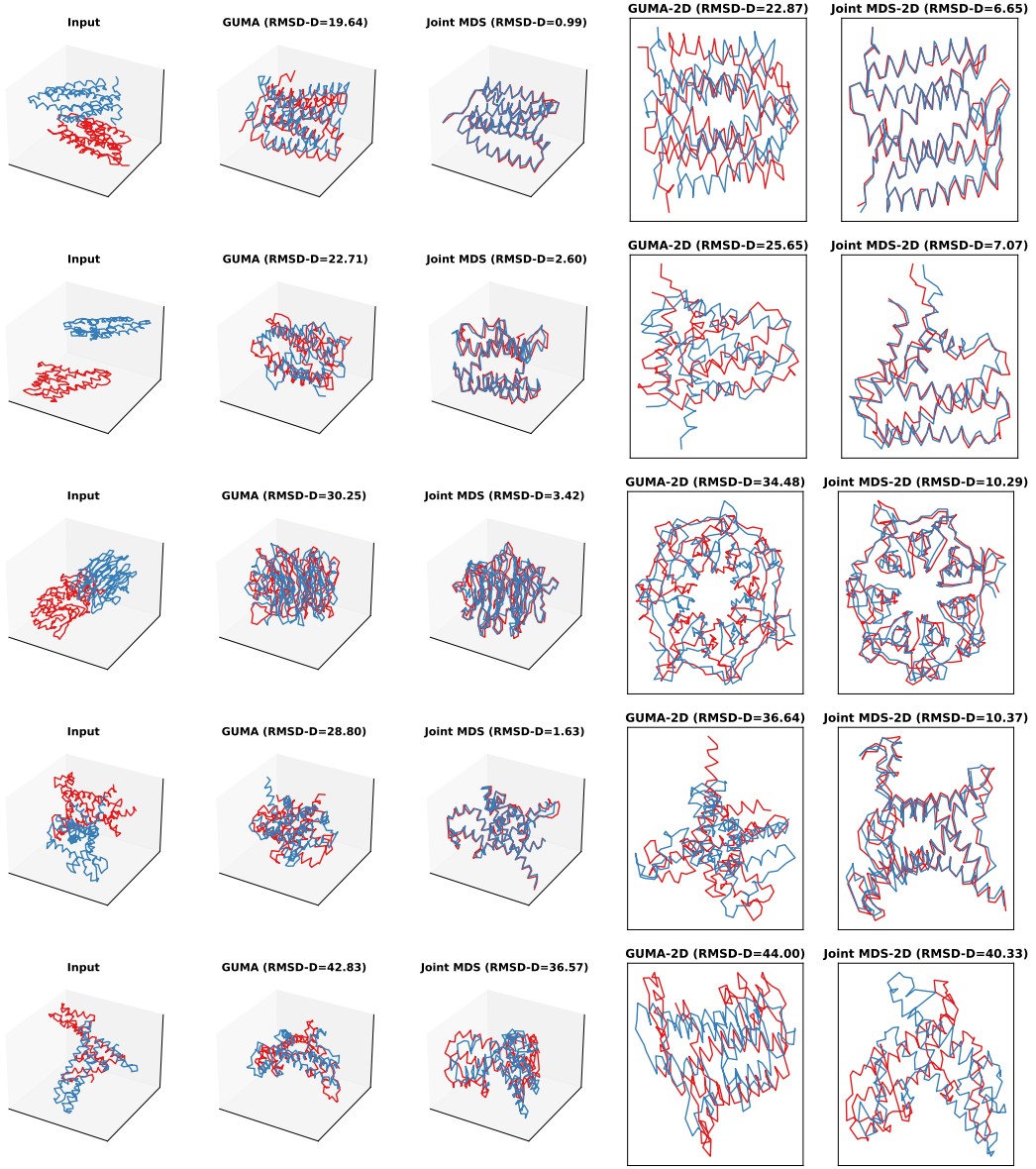

Figure 8: Comparison of GUMA and Joint MDS for protein structure alignment. The protein models for top to bottom are respectively TS1024, TS1039, TS1050, TS1058 and TS1099. Joint MDS generally outperforms GUMA except for the last case where both methods fail.

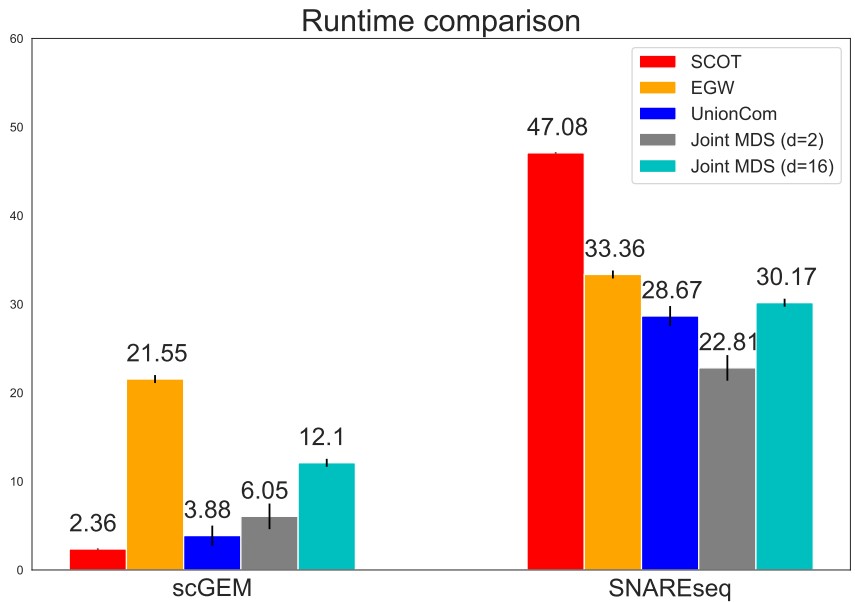

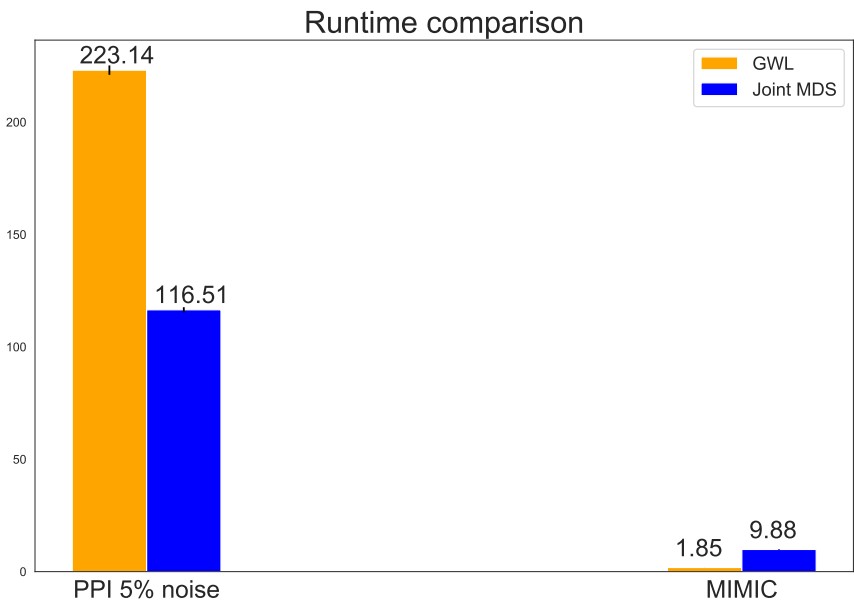

Figure 9: Runtime comparison against baseline methods for heterogeneous domain adaptation (top) and graph matching (bottom).

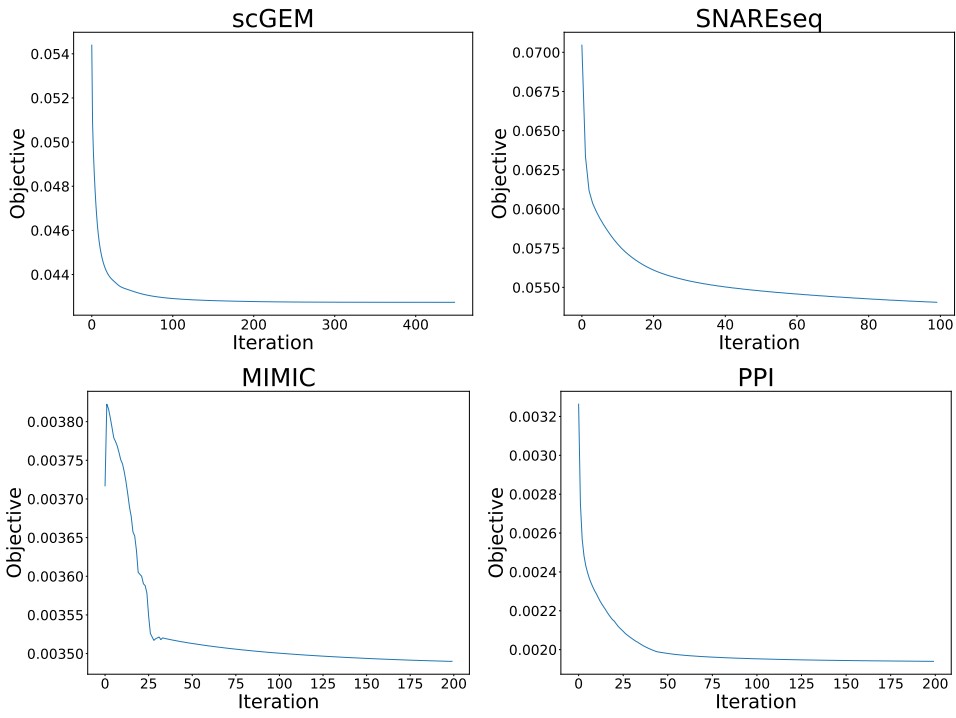

Figure 10: Convergence curves for scGEM (top left), SNAREseq (top right), MIMIC (bottom left) and PPI (bottom right).

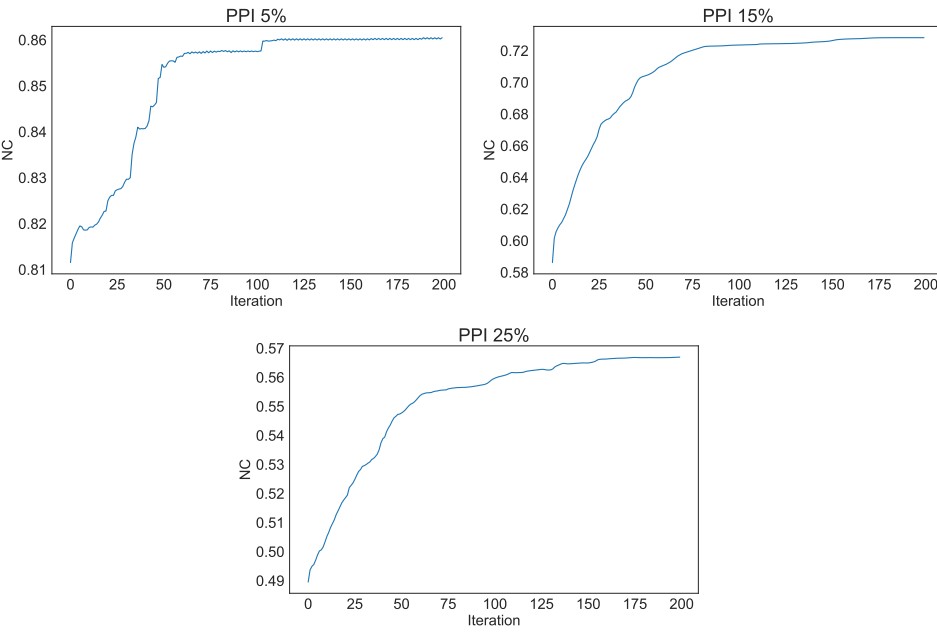

Figure 11: Node correctness with respect to number of iterations for the PPI matching task with noise level 5% (top left), 15% (top left) and 25% (bottom).

