# OpenReview forum: "Unsupervised Manifold Alignment with Joint Multidimensional Scaling"
_ICLR.cc/2023/Conference — ICLR 2023 poster_

### Official Review · Reviewer_DPfi · 2022-10-19

**Confidence:** 4
**Correctness:** 4
**Technical Novelty And Significance:** 3
**Empirical Novelty And Significance:** 3
**Recommendation:** 8

**Clarity, Quality, Novelty And Reproducibility:**

## Clarity

The article is pleasant to read and easy to understand.

## Quality

The proposed work is good.

## Novelty

While the problematic itself is not new, the proposed joint optimization is novel and may lead to new methods in different area.

## Reproducibility

The code by itself is not available, but the algorithm is well described and with the appendix a direct implementation should be possible. All the experiments are clearly described with details in appendix.

**Strength And Weaknesses:**

## Strength

The proposed framework is interesting on several points,
1. It relies on tools with good mathematical properties. Optimal transport has now effective methods for solving the problem with appropriate regularization and MDS can be approximated with the SMACOF algorithm.Thus, both sub-optimization problems are easily solved.
2. The experiments are strong with very good results for graph matching leading to potentially new methods.
3. The generality of the framework opens it to many problems that occurs in different community (especially all the problematic around matching).
4. The method seems easy to code and then enjoy a good reproducibility.

## Weaknesses

However I see several issues in the current work,
1. While the state-of-art is mostly complete, the authors forget some recent methods that may be competititve [1].
2. For graph matching, a comparison against KerGM [2] would be welcomed.
3. The method does not scale to large dataset.
4. There is no analysis of the convergence of Algorithm 1. Since it is a classical two stage optimization scheme, with appropriate hypothesis, convergence toward a stationary point should be possible [3].
5. There is no comments about the initialization. How dependent are the results to the initialization? Is it stable enough?

[1] Jin, K., Liu, C., & Xia, C. (2021). Two-Sided Wasserstein Procrustes Analysis. In IJCAI (pp. 3515-3521).
[2] Zhang, Z., Xiang, Y., Wu, L., Xue, B., & Nehorai, A. (2019). KerGM: Kernelized graph matching. Advances in Neural Information Processing Systems, 32.
[3] Bezdek, J. C., & Hathaway, R. J. (2003). Convergence of alternating optimization. Neural, Parallel & Scientific Computations, 11(4), 351-368.

**Summary Of The Paper:**

This paper proposes a new framework for unsupervised manifold alignment with joint multidimensional scaling. It combines a MDS method with a optimal transport to join the embedding. The main contribution is the combination of these two methods forming a general framework and a optimization scheme to solve both problems jointly. The experiments shows interesting results on two different tasks (visual exploration and graph matching).

**Summary Of The Review:**

This article proposes a new joint optimization problem for both multidimensional scaling with unsupervised manifold alignment. It relies on two well founded methods: multidimensional scaling  with SMACOF as solver and optimal transport with Sinkhorn's solver. The complete framework is general enough to be applied on different kind of tasks and the experiments shows promising results. Despite some weakness, this framework may interest a large part of the community.

---

> ### Author Response · Authors · 2022-11-18
> **Author response to Reviewer DPfi**
>
> Thank you for your encouraging words, helpful feedback and detailed comments, which we have tried to address below and in the revised manuscript. If you have any further questions or doubts, please let us know.
>
> > While the state-of-art is mostly complete, the authors forget some recent methods that may be competititve [1].
>
> Thank you for the reference, we have included it in the Section 2 of the revised manuscript, with a brief description.
>
> > For graph matching, a comparison against KerGM [2] would be welcomed.
>
>
> Thank you for your suggestion and we have included results for KerGM on the two graph matching tasks. We used the official implementation and the recommended parameters, which are available at [this link](https://github.com/ZhenZhang19920330/KerGM_Code/tree/master/PPInetwork).
>
> You can find below the performance comparison between KerGM and our method. KerGM achieved comparable results on the MIMIC dataset, while on the PPI datasets, our method significantly outperforms KerGM. We have also updated Table 3 in the revised manuscript accordingly.
>
> |          | PPI 5% | PPI 15% | PPI 25% | MIMIC top 3 | MIMIC top 5 |
> |----------|--------|---------|---------|-------------|-------------|
> | KerGM    | 66.14  | 39.04   | 32.17   | 22.67       | 47.86       |
> | JointMDS | 86.44  | 72.31   | 55.30   | 30.24       | 46.28       |
>
> __Note that__ the performance of KerGM in the PPI network matching task is worse than the original paper due to a bug in the original implementation that causes the input graph pairs to always be the clean PPI network and its permutation, regardless of the noise level (see [here](https://github.com/ZhenZhang19920330/KerGM_Code/blob/257813891c1faacee7f447030e21825b77974f5f/PPInetwork/main_randomwalkEdgeAtt.m#L13)). We fixed this bug and reran the experiments with the correct graph pairs.
>
>
> > The method does not scale to large dataset.
>
> As you may see from Section 4.3 and the conclusion section, we also recognize that the complexity of our method is quadratic in the number of samples (its complexity is $O(T(d^3+2N^2))$ where $T$ is the total number of iterations, $d$ is the embedding dimensions and $N$ is the total number of samples, as stated in Section 4.3), which is the major bottleneck for its application to large-scale datasets. We would also like to point out that extending ML concepts to a large-scale setting is typically a project of its own and many ideas are presented first in a non-scalable variant. We definitely aim to accelerate our method in future work, and one possible direction would be to reduce the computation cost of MDS to linear complexity through the lens of the recent advancement in stochastic MDS.
>
> > There is no analysis of the convergence of Algorithm 1. Since it is a classical two stage optimization scheme, with appropriate hypothesis, convergence toward a stationary point should be possible [3].
>
> Convergence analysis of non-convex optimization is very difficult in general. There are very few known theoretical results on the convergence of alternating optimization for non-convex problems. Moreover, even for the Wasserstein Procrutes problem, we are not aware of any work (e.g. Alvarez-Melis et al., 2019; Grave et al., 2019) that mathematically proves the convergence of their algorithms. Thus, we leave the theoretical analysis of convergence for future work. Despite this theoretical challenge, we empirically find that our method converges stably on different tasks and datasets, as shown in __Figure 10__ in the Appendix.
>
> > There is no comments about the initialization. How dependent are the results to the initialization? Is it stable enough?
>
> Initialization is important for the Wasserstein Procrutes problem, especially for the the coupling matrix $\mathbf{P}$, as you may see from the Section 4 about the __overall algorithm__. In the low-dimensional case (for the application of joint visualization of two datasets and protein structure alignment), we note that Initializing the embeddings $\mathbf{Z}$ and $\mathbf{Z}'$ with two individual SMACOF algorithms works already well and stably. However in the high-dimensional case, GW could be used as a convex relaxation of the Wasserstein Procrustes problem (Alvarez-Melis et al., 2019; Grave et al., 2019), which provides a better initialization for the coupling matrix $\mathbf{P}$. Even though the coupling matrix $\mathbf{P}$ is intialized with GW (of already a not bad quality), its quality further improves with the number of iterations of Joint MDS, thanks to regularization through explicit learning of embeddings. To demonstrate this, we have included the curves for accuracy of graph matching versus the number iterations in __Figure 11__ of the Appendix in the revised manuscript.
>
> > The code by itself is not available, but the algorithm is well described and with the appendix a direct implementation should be possible.
>
> Please note that in fact we did attach the code for all experiments in the supplementary file.

---

### Official Review · Reviewer_hbpW · 2022-10-23

**Confidence:** 5
**Correctness:** 4
**Technical Novelty And Significance:** 3
**Empirical Novelty And Significance:** 2
**Recommendation:** 3

**Clarity, Quality, Novelty And Reproducibility:**

The paper is clearly written. While the proposed extension of the classical MDS technique enables new possibilities for UDA, the authors have chosen rather simple datasets to demonstrate the results of their approach.

**Strength And Weaknesses:**

Strengths: A useful extension of MDS with applications in unsupervised domain adaptation (UDA) is proposed. This approach enables the mapping of datasets from two different domains onto a latent space.
Weaknesses: I am not impressed with papers that have reasonable theory but demonstrate results on synthetic data, MNIST etc. Results in table 3 are not impressive either.


**Summary Of The Paper:**

This paper generalizes the classical  multidimensional scaling approach for unsupervised manifold alignment. This enables the mapping of , datasets from two different domains, without requiring correspondences across the datasets, to a common low-dimensional Euclidean space. This approach generalizes approaches based on dictionaries that map the source and target data onto a common latent space. Results on a few benchmark datasets are proposed.

**Summary Of The Review:**

I will encourage the authors to do more experiments with real and challenging datasets.

---

> ### Author Response · Authors · 2022-11-18
> **Author response to Reviewer hbpW**
>
> Thank you for your time and comments. Please find below our response.
>
> > While the proposed extension of the classical MDS technique enables new possibilities for UDA, the authors have chosen rather simple datasets to demonstrate the results of their approach. I will encourage the authors to do more experiments with real and challenging datasets.
>
> Regarding your comment, please note the following points:
>
> - First, our method is not limited to UDA. We address the general problem of unsupervised manifold alignment with few assumptions on data modality, i.e., only intra-dataset pairwise dissimilarities are required. This more general setting, which differs from previous work, enables our method to be used in a variety of applications. As a result, UDA is only one of the five applications considered in this work. The goal of our experiments is not to achieve state-of-the-art results for all these tasks but to demonstrate that JointMDS is a versatile analysis tool applicable to a variety of machine learning tasks fitting into our general setting. In addition, we believe that most existing methods developed for UDA cannot be trivially applied to the other tasks considered in this work, such as graph matching and protein structure alignment.
> -  Second, we do evaluate our method on real-world datasets. Specifically, the PPI and MIMIC datasets that we used for graph matching were also widely used in recently studies such as [1-3]. The scGEM and SNAREseq datasets that we used for visualization and UDA are also real-world single-cell multi-omics data recently provided by Demetci et al. 2022. Protein structure alignment dataset is constructed from the CASP14 database, a very recent database created in 2020 that have been used to benchmark many recently developed protein structure prediction methods, including AlphaFold2. Our method performs well on all of these datasets, showing its potential applicability to a variety of machine learning problems.
>
> [1] Xu, Hongteng, Dixin Luo, and Lawrence Carin. "Scalable gromov-wasserstein learning for graph partitioning and matching." Advances in neural information processing systems (NeurIPS), 2019.
>
> [2] Xu, Hongteng, et al. "Gromov-wasserstein learning for graph matching and node embedding." International conference on machine learning (ICML), 2019.
>
> [3] Zhang, Zhen, et al. "KerGM: Kernelized graph matching." Advances in Neural Information Processing Systems 32 (2019).

---

### Official Review · Reviewer_jgCM · 2022-10-24

**Confidence:** 3
**Correctness:** 3
**Technical Novelty And Significance:** 2
**Empirical Novelty And Significance:** 2
**Recommendation:** 6

**Clarity, Quality, Novelty And Reproducibility:**

This work can be considered as marginal contribution on existing work from MDS optimization and Wasserstein Procrustes analysis.

Author provides the code and data in the supplementary material; this is good to see.

Also, given sequence alignment has been considered in this work, it seems that the reference to the previous spatial-temporal alignment domain (i.e., which is also join alignment between 2 different data sets with some intrinsic structure, see reference 2 below) can be added, for the common and differences in the framework.

Reference:
2. F. Zhou and F. De la Torre, Canonical Time Warping for Alignment of Human Behavior, Advances in Neural Information Processing Systems (NeurIPS), 2009





**Strength And Weaknesses:**

Strengths:

The problem itself is interesting and the article is pleasant to read too, i.e., join alignment and embedding from 2 different data sets without supervision can be find a lot of application across NLP, computer vision, healthcare, biology, etc.

The optimization part is also solid as with enough considerations of previous works to joint optimize this non-convex problem by SMACOF algorithm and Wasserstein optimizations from (Alvarez-Melis et al., 2019), and with detailed analysis of computational complexity (Section 4.3), which is quite helpful to see.

Experimental results also can be considered solid as across a number of synthetic or real data coupled with different tasks (with more results in Appendix B), and the proposed method often show (not always) the relative best performance among existing methods.

Weaknesses:

One of the key concerns is, the proposed joint MDS method is based on MDS as the starting point, which utilize the (dis) similarities from Euclidean space and this may conflict with the nature of manifold embedding when data with strong nonlinear intrinsic geometry structure. Indeed, there are some discussions provided in Section 4.2, which is good to see, but still some unclear remains. As ISOMAP is essentially MDS coupled with geodesic distance, and not sure this is applied in which part of results in Section 5? (Maybe I missed).  Also, outside MDS and ISOMAP, there are other graph spectral works in this direction such as Laplacian Eigenmaps (and others), should be helpful to have some judgment for the choice in this work.

Additionally, it seems using GW as the initialization step (mentioned in Section 4.3) to provide a better coupling matrix P in the algorithm, make the contribution from this proposed iterative optimization less clear as maybe the non-convex formulation is good in theory but less practical in the numerical way.

Reference:
1. D. Alvarez-Melis, S. Jegelka, and T. S Jaakkola. Towards optimal transport with global invariances. In International Conference on Artificial Intelligence and Statistics (AISTATS), 2019

**Summary Of The Paper:**

This work proposes a practical method to jointly estimate the low dimension embedding in Euclidean space and the correspondences between data sets, which can be viewed as the alignment extension of classical embedding work by Multidimensional Scaling (MDS). Unlike some previous works in this domain, the proposed method is unsupervised by nature without given any known subset of alignment.

This joint MDS framework is formulated as a combined stress objective function plus the additional matching penalty term from Wasserstein Procrustes analysis. Given this proposed join objective function is non-convex (same as similar problems), this paper proposed an alternative optimization algorithm (similar to EM type), which can be viewed as the combination of weighted MDS problem and Wasserstein Procrustes problem in each iteration.

Experimental results on both synthetic data and real-world data (e.g., cell, MNIST, human body data, protein, etc) on a number of tasks from joint 2D visualization, domain adaptation, graph matching and protein alignment.

**Summary Of The Review:**

The overall contribution seems marginal but given the solid technical work behind this alternative optimization algorithm, and the rich empirical evaluation to this joint alignment problem, my recommendation for this paper is 6 "marginally above the acceptance threshold".

---

> ### Author Response · Authors · 2022-11-18
> **Author response to Reviewer jgCM**
>
> Thank you for your time and detailed comments. We have addressed the points below and in the revised manuscript. If you have further questions, please let us know.
>
> > As ISOMAP is essentially MDS coupled with geodesic distance, and not sure this is applied in which part of results in Section 5? (Maybe I missed).
>
> We indeed used geodesic distances for joint visual exploration of two datasets and unsupervised heterogeneous domain adaptation (Section 5.1 and 5.2). Note that for graph matching, we naturally used the geodesic distance on the graph as no data features are available. This was described in a sentence in the experimental setup paragraph in Section 5.1: "We compute the pairwise geodesic distances of each dataset as described in 4.2." The choice of the hyperparameters, such as the number of nearest neighbors to construct the graph for each dataset, is detailed in Section B.2.2 of the Appendix.
>
> > Also, outside MDS and ISOMAP, there are other graph spectral works in this direction such as Laplacian Eigenmaps (and others), should be helpful to have some judgment for the choice in this work.
>
> Thank you for the suggestion. Laplacian Eigenmaps could indeed be an interesting alternative to MDS. We chose the MDS because it does not impose any constraints on the embeddings, while Laplacian Eigenmaps force the embeddings to have a unit norm for each coordinate vector, which could be harder to optimize. We leave the adaptation of our method to Laplacian Eigenmaps for future work.
>
> > it seems using GW as the initialization step (mentioned in Section 4.3) to provide a better coupling matrix P in the algorithm, make the contribution from this proposed iterative optimization less clear as maybe the non-convex formulation is good in theory but less practical in the numerical way.
>
> Initialization is important for the Wasserstein Procrutes problem, especially for the the coupling matrix $\mathbf{P}$. In the low-dimensional case (for the application of joint visualization of two datasets and protein structure alignment), we note that Initializing the embeddings $\mathbf{Z}$ and $\mathbf{Z}'$ with two individual SMACOF algorithms works already well. However in the high-dimensional case, GW could be used as a convex relaxation of the Wasserstein Procrustes problem (Alvarez-Melis et al., 2019; Grave et al., 2019), which provides a better initialization for the coupling matrix $\mathbf{P}$. Even though the coupling matrix $\mathbf{P}$ is intialized with GW (of already a not bad quality), its quality further improves with the number of iterations of Joint MDS, thanks to regularization through explicit learning of embeddings. To demonstrate this, we have included the curves for accuracy of graph matching versus the number iterations in __Figure 11__ of the Appendix in the revised manuscript.
>
> > given sequence alignment has been considered in this work, it seems that the reference to the previous spatial-temporal alignment domain (i.e., which is also join alignment between 2 different data sets with some intrinsic structure, see reference 2 below) can be added, for the common and differences in the framework.
>
> Thank you for the reference. Nevertheless, the task we considered in this work is protein structure alignment which aligns two sets of 3D coordinates, rather than sequence alignment.

---

### Official Review · Reviewer_rQ9i · 2022-10-25

**Confidence:** 4
**Correctness:** 4
**Technical Novelty And Significance:** 2
**Empirical Novelty And Significance:** 2
**Recommendation:** 6

**Clarity, Quality, Novelty And Reproducibility:**

Clarity: The paper is clearly written and easy to follow.

Novelty: As I commented, the novelty of this work seems limited.

**Strength And Weaknesses:**

Strength:

1. The paper is very well-written and technically sound. The idea of the paper is natural and clearly motivated.

2. The proposed method is effective on a variety of datasets.

Weakness and questions:

1. The novelty of the paper seems limited to me. It looks to me that this work simply combines the methods in two regimes and implement a joint optimization.

2. From the optimization perspective, the proposed method is essentially solving the two problems (MDS and Wasserstein Procrustes) in an alternating way. It is hard to tell whether the resulting heuristic algorithm will lead to convergence to a local/global min.



**Summary Of The Paper:**

This paper proposes to embed two sets of data into a common low dimensional space and at the same time estimate the correspondences between the data points of two sets.  The proposed method combines Multidimensional Scaling (MDS) and Wasserstein Procrustes analysis into a joint optimization problem, which is solved using alternating updates. The authors demonstrate the effectiveness of the proposed method on a variety of datasets.

**Summary Of The Review:**

In summary, the idea of the paper is very natural for solving the proposed task, although its novelty looks limited. In general, I enjoyed reading the paper. Given both of its limitations and advantages, this work has a potential to be published at ICLR.

---

> ### Author Response · Authors · 2022-11-18
> **Author response to Reviewer rQ9i**
>
> Thank you for your time and helpful comments. We have addressed the concerns below and in the revised manuscript. If you have any further questions, please let us know.
>
> > The novelty of the paper seems limited to me. It looks to me that this work simply combines the methods in two regimes and implement a joint optimization.
>
> In this work, we address the general problem of unsupervised manifold alignment with few assumptions on data modality, i.e., only intra-dataset pairwise dissimilarities are required. This more general setting, which differs from previous work, is primarily motivated by applications where the data does not have access to the input features, such as graph matching. The novelty of our work lies in the fact that we formulate this interesting machine learning problem as a new optimization problem and propose a solution to this problem. The proposed solution can freely benefit from the techniques developed for solving each sub-problem, namely MDS and Wasserstein Procrustes. We demonstrate the effectiveness of our method in a variety of machine learning applications in a unified and systematic way. We believe our formulation is original and the proposed solution could be applied in many fields and can be reused and adapted in the future research on manifold alignment and optimal transport.
>
> > From the optimization perspective, the proposed method is essentially solving the two problems (MDS and Wasserstein Procrustes) in an alternating way. It is hard to tell whether the resulting heuristic algorithm will lead to convergence to a local/global min.
>
> Convergence analysis of non-convex optimization is very difficult in general. There are very few known theoretical results on the convergence of alternating optimization for non-convex problems. Moreover, even for the Wasserstein Procrutes problem, we are not aware of any work (e.g. Alvarez-Melis et al., 2019; Grave et al., 2019) that proves the convergence of their algorithms. Thus, we leave the theoretical analysis of convergence for future work. Note that despite this theoretical challenge, we empirically find that our method converges stably on different tasks and datasets, as shown in __Figure 10__ in the Appendix.

---

### Author Response · Authors · 2022-11-18
**General comments**

Dear reviewers,

We would like to thank you for your time and effort in providing such helpful and detailed comments, which we believe have strengthened our paper considerably. We will address specific comments and concerns in the individual reviews (and also in the revised manuscript, where the changes can be seen in red). We thank you for your positive feedback:

- (__Reviewer rQ9i, hbpW, jgCM__) The paper is well written and easy to follow, with the idea being clearly motivated.
- (__Reviewer jgCM, DPfi__) The problem that the paper aimed to tackle is very interesting and has great potential to be extended to many different research fields.
- (__Reviewer rQ9i, jgCM, DPfi__) The proposed method is technically/mathematically sound and the solid experiments have shown its very good performance in various applications, with very good results for graph matching.
- (__Reviewer jgCM, DPfi__) The work is reproducible with the code available for all experiments.

---

> ### Comment · Reviewer_DPfi · 2022-11-30
> **About the final version of the paper and the comments**
>
> Many thanks for your answer and updates of the paper. I find this work really interesting as graph matching is complex.The current methods based on optimal transport have all their pro and cons, and I think this works solve some of pros (mostly about optimization and stability). Moreover the proposed method shows a pretty good robustness to noise, even better than KerGM and this is really important for real data. I stay with my score.

---

> > ### Author Response · Authors · 2022-12-02
> > **Thank you**
> >
> > Thank you for considering our rebuttal and manuscript revision in a timely manner, and for the encouraging words. We will be monitoring the thread if any other questions arise.

---

### Decision · Program_Chairs · 2023-01-20

**Decision:**

Accept: poster

**Justification For Why Not Higher Score:**

This idea is simple and it is good to have a method that combines MDS and Wasserstein distance. Many reviewers are happy to accept. However, they are not that excited about the results. At this point, I lean towards acceptance, however, it is completely fine that it can be rejected.

**Justification For Why Not Lower Score:**

This idea is simple and it is good to have a method that combines MDS and Wasserstein distance. Many reviewers are happy to accept.

**Metareview: Summary, Strengths And Weaknesses:**

In this paper, the authors propose a new manifold alignment, which jointly estimate the low dimension embedding in Euclidean space and the correspondences between data. More specifically, the proposed method consists of Multidimensional Scaling (MDS) and Wasserstein Procrustes analysis, and the model parameters are updated in an alternating manner. Through experiments, the authors demonstrated that the proposed method compares favorably with existing methods.

The proposed method is simple yet effective. However, the technical contribution is a bit limited; this is a borderline paper.

**Note From Pc:**

if the above contains the word "oral" or "spotlight" please see: "oral" presentation means -> notable-top-5% and "spotlight" means -> notable-top-25%. As stated in our emails, we are disassociating presentation type from AC recommendations

**Summary Of Ac-Reviewer Meeting:**

N/A